# Merging organoid and organ-on-a-chip technology to generate complex multi-layer tissue models in a human retina-on-a-chip platform

Kevin Achberger[1†], Christopher Probst[2†], Jasmin Haderspeck[1†], Sylvia Bolz[3], Julia Rogal[2,4], Johanna Chuchuy[2,4], Marina Nikolova[1], Virginia Cora[1], Lena Antkowiak[1], Wadood Haq[3], Nian Shen[4], Katja Schenke-Layland[4,5,6], Marius Ueffing[3], Stefan Liebau[1]*, Peter Loskill[2,4]*

[1]Institute of Neuroanatomy & Developmental Biology (INDB), Eberhard Karls University Tübingen, Tübingen, Germany; [2]Fraunhofer Institute for Interfacial Engineering and Biotechnology IGB, Stuttgart, Germany; [3]Centre for Ophthalmology, Institute for Ophthalmic Research, Eberhard Karls University Tübingen, Tübingen, Germany; [4]Department of Women's Health, Research Institute for Women's Health, Eberhard Karls University Tübingen, Tübingen, Germany; [5]Natural and Medical Sciences Institute (NMI), Reutlingen, Germany; [6]Department of Medicine/Cardiology, Cardiovascular Research Laboratories, David Geffen School of Medicine, Los Angeles, United States

*For correspondence:
stefan.liebau@uni-tuebingen.de (SL);
peter.loskill@igb.fraunhofer.de (PL)

†These authors contributed equally to this work

Competing interests: The authors declare that no competing interests exist.

**Abstract** The devastating effects and incurable nature of hereditary and sporadic retinal diseases such as Stargardt disease, age-related macular degeneration or retinitis pigmentosa urgently require the development of new therapeutic strategies. Additionally, a high prevalence of retinal toxicities is becoming more and more an issue of novel targeted therapeutic agents. Ophthalmologic drug development, to date, largely relies on animal models, which often do not provide results that are translatable to human patients. Hence, the establishment of sophisticated human tissue-based in vitro models is of upmost importance. The discovery of self-forming retinal organoids (ROs) derived from human embryonic stem cells (hESCs) or human induced pluripotent stem cells (hiPSCs) is a promising approach to model the complex stratified retinal tissue. Yet, ROs lack vascularization and cannot recapitulate the important physiological interactions of matured photoreceptors and the retinal pigment epithelium (RPE). In this study, we present the retina-on-a-chip (RoC), a novel microphysiological model of the human retina integrating more than seven different essential retinal cell types derived from hiPSCs. It provides vasculature-like perfusion and enables, for the first time, the recapitulation of the interaction of mature photoreceptor segments with RPE in vitro. We show that this interaction enhances the formation of outer segment-like structures and the establishment of in vivo-like physiological processes such as outer segment phagocytosis and calcium dynamics. In addition, we demonstrate the applicability of the RoC for drug testing, by reproducing the retinopathic side-effects of the anti-malaria drug chloroquine and the antibiotic gentamicin. The developed hiPSC-based RoC has the potential to promote drug development and provide new insights into the underlying pathology of retinal diseases.
DOI: https://doi.org/10.7554/eLife.46188.001

## Introduction

Retinal diseases such as Stargardt disease, age-related macular degeneration, diabetic retinopathies or retinitis pigmentosa are amongst the leading causes of vision loss in humans (*Croze et al., 2014*; *Buch et al., 2004*). Unfortunate for patients suffering from those diseases, there are currently no cures available (*Shintani et al., 2009*; *Fine et al., 2000*). Moreover, the complex neuro-retinal organization and the vast blood-supply make retinal tissue susceptible for side effects of compounds delivered intravitreally or systemically (*Penha et al., 2010*; *Renouf et al., 2012*). Retinal toxicities are a major issue for a wide range of therapeutic substances, especially for targeted anticancer agents since many of the targets are also expressed in ocular tissues (*Renouf et al., 2012*). Although animal models that are used to explore new therapeutic options and assess retinal toxicities resemble the human (patho-)physiology of vision in certain aspects, they fail to reflect fundamental characteristics including trichromacy or a fovea centralis, responsible for high visual acuity (*Figure 1*). In vitro cell culture assays, on the other hand, are typically based on non-physiological 2D cell cultures, which cannot reflect the complex architecture and cell-cell interactions as well as the blood perfusion. More complex approaches such as retinal explants from human donors provide a full-featured model; however, the limited availability and culturability as well as inter-donor variabilities make it unsuited for drug development and testing. The invention of physiologically relevant in vitro models

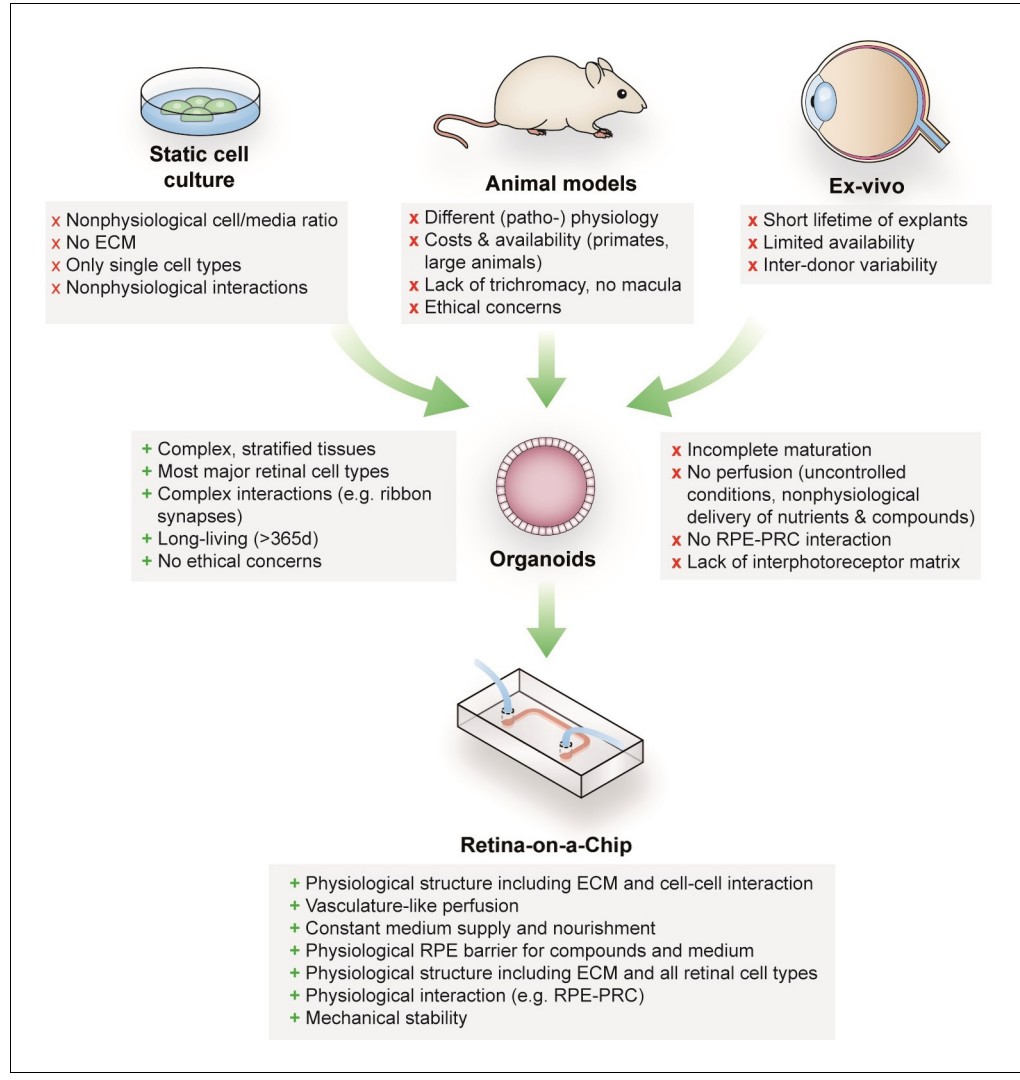

**Figure 1.** Advantages and limitations of retinal models for drug development and mechanistic research.
DOI: https://doi.org/10.7554/eLife.46188.002

capable of mimicking the human retinal biology is hence of crucial importance. Recent progress in the generation of 3-dimensional (3D) organoids derived from human pluripotent stem cells (hPSC) (derived from both induced (hiPSC) as well as embryonic (hESC) stem cells) enabled the reflection of distinct types of tissues, such as subsystems of the central nervous system including the retina. Retinal organoids (ROs), also called 'eyes in the dish', resemble rudimentary optic vesicle-like structures with a retinal layering similar to in vivo conditions (*Nakano et al., 2012*; *Zhong et al., 2014*). These ROs contain most relevant retinal cell types in a physiological layering such as ganglion cells, amacrine cells, horizontal cells, bipolar cells, Müller glia as well as rods and cones. Nevertheless, hPSC-ROs are still facing a variety of drawbacks limiting predictive research on for example human retinal development, function or drug response. Some of the major hurdles are (i) the functional maturation of differentiated cells, (ii) lack of essential cell types (e.g. microglia), (iii) lack of a physiological interplay of the various retinal cell types especially of photoreceptors and retinal pigment epithelia (RPE), as well as (iv) a missing vascularization (reviewed in *Achberger et al., 2019*; *Yin et al., 2016*). Due to the lack of a physiological perfusion, the delivery of compounds to ROs is uncontrolled and entirely artificial. Here, general limitations of static cell culture apply including non-physiological cell-to-media ratio, uncontrolled shear forces during media exchanges, as well as highly variable conditions between media exchanges. In recent years, the short-comings of conventional static cell culture has led to the emergence of microphysiological systems (MPS), specifically Organ-on-a-Chip (OoC) platforms. MPSs have evolved into a powerful alternative for classical cell culture and animal models by providing physiological microenvironments embedded in a vascular-like microfluidic perfusion (reviewed in *Wikswo, 2014*; *Zhang et al., 2018*). This new and promising technology has the potential to revolutionize drug development and usher into a new era of personalized medicine. Over the past years, a variety of MPSs have been developed, mimicking, for instance, cardiac (*Mathur et al., 2015*; *Agarwal et al., 2013*), lung (*Huh et al., 2010*), renal (*Wilmer et al., 2016*), and hepatic tissue (*Bhise et al., 2016*; *Nakao et al., 2011*). In the context of ophthalmologic research, a variety of approaches have been introduced that represent partial layers of the cornea (*Puleo et al., 2009*) or the retina (*Chen et al., 2017*; *Dodson et al., 2015*; *Su et al., 2015*; *Jeon et al., 2016*; *Mishra et al., 2015*). So far, however, no MPS has been able to successfully recapitulate the complex 3D architecture of the human retina.

In this study, we developed a physiologically relevant 3D in vitro model of the human retina by combining hiPSC-ROs with hiPSC-derived RPE in a retina-on-a-chip (RoC). This novel microphysiological platform enables enhanced inner and outer segment formation and preservation, a direct interplay between RPE and photoreceptors as well as a precisely controllable vasculature-like perfusion. In order to provide a high-content platform for basic and applied research, we established a toolbox comprising in situ analysis approaches as well as terminal endpoints enabling the monitoring of functionality as well as molecular mechanisms. To demonstrate the applicability for drug screening, the system was exposed to the drugs chloroquine and gentamicin, which are known to have retinopathic side effects (*Elman et al., 1976*; *Ding et al., 2016*; *Yusuf et al., 2017*; *Zemel et al., 1995*; *McDonald et al., 1986*).

## Results

### Retinal organoids show rod and cone diversity and simple inner and outer segment formation

ROs derived from hiPSCs harbor all known major retinal subtypes such as ganglion cells, bipolar cells, horizontal cells, amacrine cells, Müller glia and photoreceptors (*Zhong et al., 2014*; *Figure 2—figure supplement 1*). Using immunostaining and gene expression analysis, the presence of the retinal cell types, as well as crucial retinal morphological cues such as inner and outer photoreceptor segment formation, a tightly formed outer limiting membrane (OLM) and a correct layering, was verified (*Figure 2—figure supplement 1*). In order to reach a suitable maturation, ROs were differentiated for 180 days. ROs of that age harbor matured photoreceptors which have simple forms of inner and outer segments, situated on the surface of the ROs, visible in bright field microscopy or by immunostaining of respective markers (*Figure 2b–f*). Immunostaining of the respective organoids demonstrates the presence of a mixed population of rods and cones, identified by specific markers (rhodopsin for rods and Arrestin-3 for cones, *Figure 2c–e*, *Figure 2—figure supplement 1*). In order

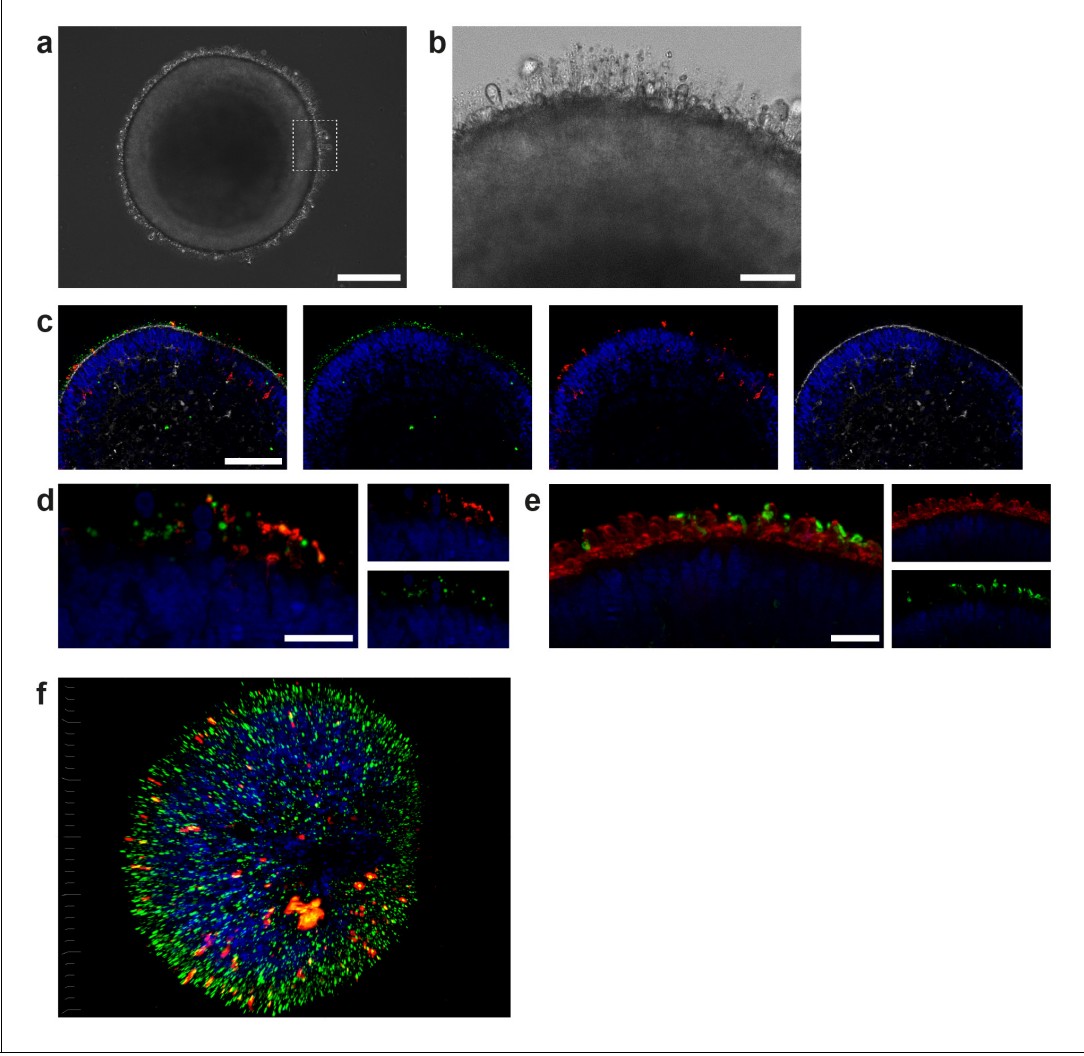

**Figure 2.** Characterization of retinal organoids. (**a**) Brightfield image of a day 180 RO in dish culture. (**b**) Magnified area of a) highlighting inner and outer segment-like structures. (**c**) Day 180 ROs cryosectioned and immunostained for the rod marker rhodopsin (red), the outer segment marker ROM1 (green) and phalloidin (white) visualizing the outer limiting membrane. (**d**) Day 180 ROs sectioned and immunostained for the rod marker rhodopsin (red) and the outer segment marker ROM1 (green). (**e**) Day 180 ROs sectioned and immunostained for the rod marker rhodopsin (green) and for PNA lectin (red). (**f**) 3D visualization of whole-mount staining of day 180 RO stained for rhodopsin (red) and ROM1 (green). Bars indicate a) 250 µm b) 50 µm c) 100 µm d-f) 20 µm. Blue: DAPI.

DOI: https://doi.org/10.7554/eLife.46188.003

The following figure supplement is available for figure 2:

**Figure supplement 1.** Cell types in dish cultured hiPSC-derived retinal organoids.

DOI: https://doi.org/10.7554/eLife.46188.004

to analyze segment formation, we used ROM1 as it has been previously shown as specific outer segment marker (*Datta et al., 2015*) and verified the segment specificity by co-staining with the rod marker rhodopsin (*Figure 2c–d,f*). Further, we tested the previously described protein PNA lectin, which was delineated to specifically bind to cone photoreceptor segments (*Blanks and Johnson, 1984*). Co-staining of rhodopsin with PNA lectin exhibited that not only cone but also rod segments in ROs are labeled with PNA lectin (*Figure 2e*).

## Microphysiological retina-on-a-chip

To recapitulate the complex in vivo anatomy of the human retina in vitro (*Figure 3a*), we developed a microfluidic platform that enables the culture of hiPSC-derived RPE and ROs in a defined

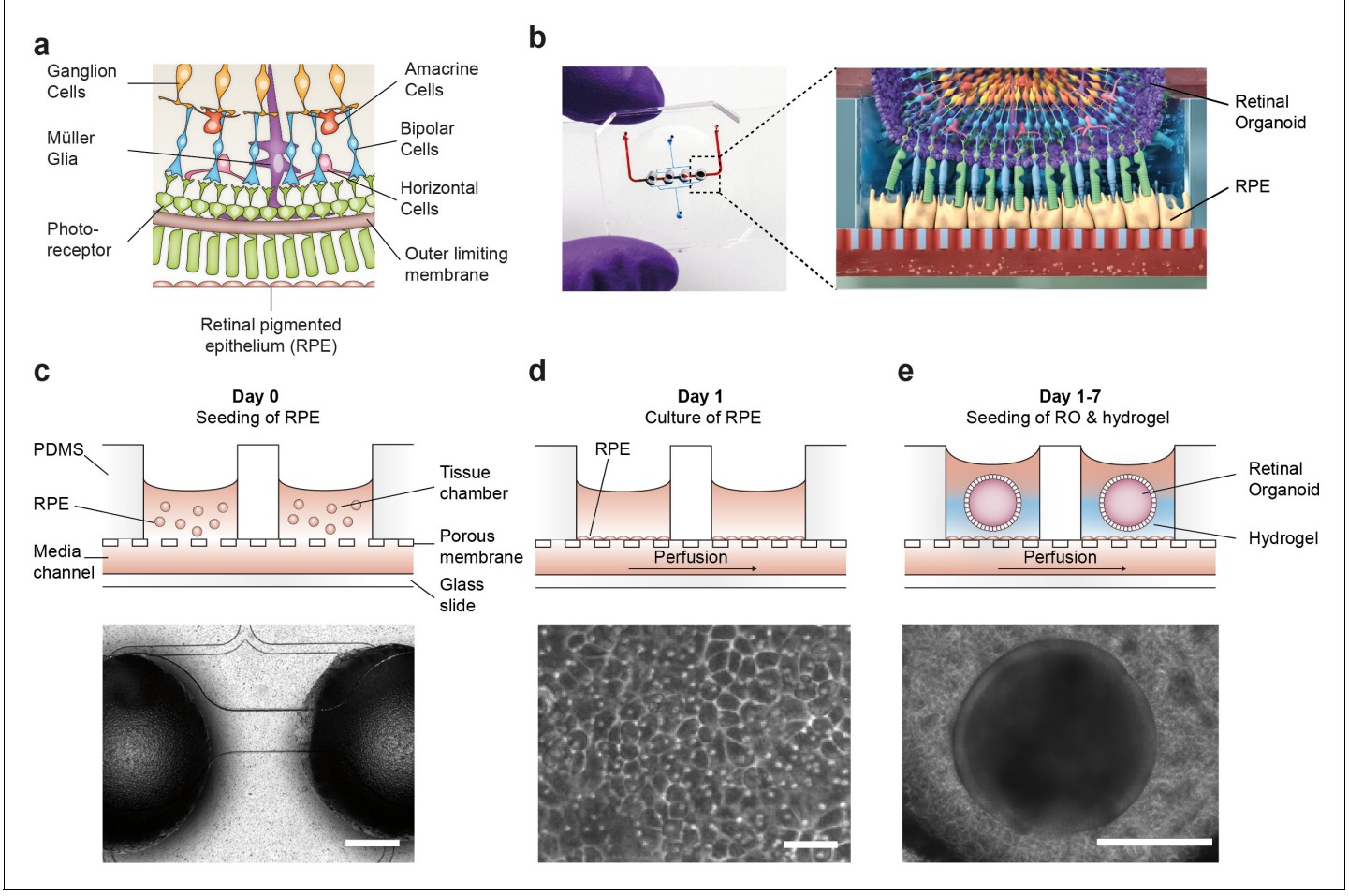

**Figure 3.** Microfluidic RoC. (**a**) Schematic representation of the human retinal composition and cell types in vivo. (**b**) Photo (left) of the RoC and (right) representation of the RO photoreceptor and RPE interaction. (**c**) RPE cells are seeded into the device, (**d**) forming a densely packed monolayer after 24 hr of culture. (**e**) ROs and the hyaluronic acid-based hydrogel are directly loaded from the top into the well and onto the RPE. Bars indicate (**c**) 500 µm, (**d**) 80 µm, (**e**) 400 µm.

DOI: https://doi.org/10.7554/eLife.46188.005

The following figure supplement is available for figure 3:

**Figure supplement 1.** Long-term culture of RoC.

DOI: https://doi.org/10.7554/eLife.46188.006

physiological structure (*Figure 2b*). Each RoC features four identical micro-tissues connected via a microchannel and is comprised of two transparent and biocompatible polymer layers. The top layer features the compartments for the ROs and RPE, whereas the bottom layer provides a channel for a vasculature-like perfusion enabling a constant supply of nutrients and compounds. Both layers are separated by a thin porous membrane mimicking the endothelial barrier and shielding the tissues from shear forces while simultaneously enabling the exchange of nutrients and metabolites (*Figure 3b*). The tissue compartments are accessible from above for the initial seeding process and sealed during the subsequent culture process to avoid evaporation and contamination. A stable tissue comprising ROs and RPE was achieved by first seeding hiPSC-derived RPE cells at a defined density into each tissue compartment (*Figure 3c*) and subsequent culture for 24 hr (*Figure 3d*). This step was followed by injection of ROs embedded in a hyaluronic acid-based hydrogel (representing the major component of the interphotoreceptor matrix between RPE and PRC) into the tissue compartments (*Figure 3e*). This led to the formation of a thin hydrogel layer, generating a defined distance between RPE cells and the outer limiting membrane of the ROs. Thereby, a direct contact

and, thus, an uncontrolled outgrowth of cells from the ROs during culture was successfully avoided. ROs and RPE were cultured in the system for at least 3 days prior to further functionality assessment or experimentation. The controlled culture conditions enabled a stable culture of the RoC for at least 21 days (*Figure 3—figure supplement 1*).

## Specific marker expression and polarization of retinal pigment epithelial cells in the RoC

A polarized and functional RPE is crucial for the survival of photoreceptors in vivo and a vital part of the visual cycle shuttling retinoids between the RPE and photoreceptor outer segments (*Kevany and Palczewski, 2010*; *Marmorstein, 2001*). Therefore, the RPE in the RoC was thoroughly tested for its marker expression and polarization (*Figure 4*, *Figure 4—figure supplement 1*). Expression of RPE markers PAX6 and MITF can be observed after 7–14 days in the chip (*Figure 4a–b*). Mature RPE inside the chip displayed cobble stone-like morphology and tight-junction formation visualized by ZO-1 staining (*Figure 4c*). The melanosome and pigmentation marker Melanoma gp100 (also called

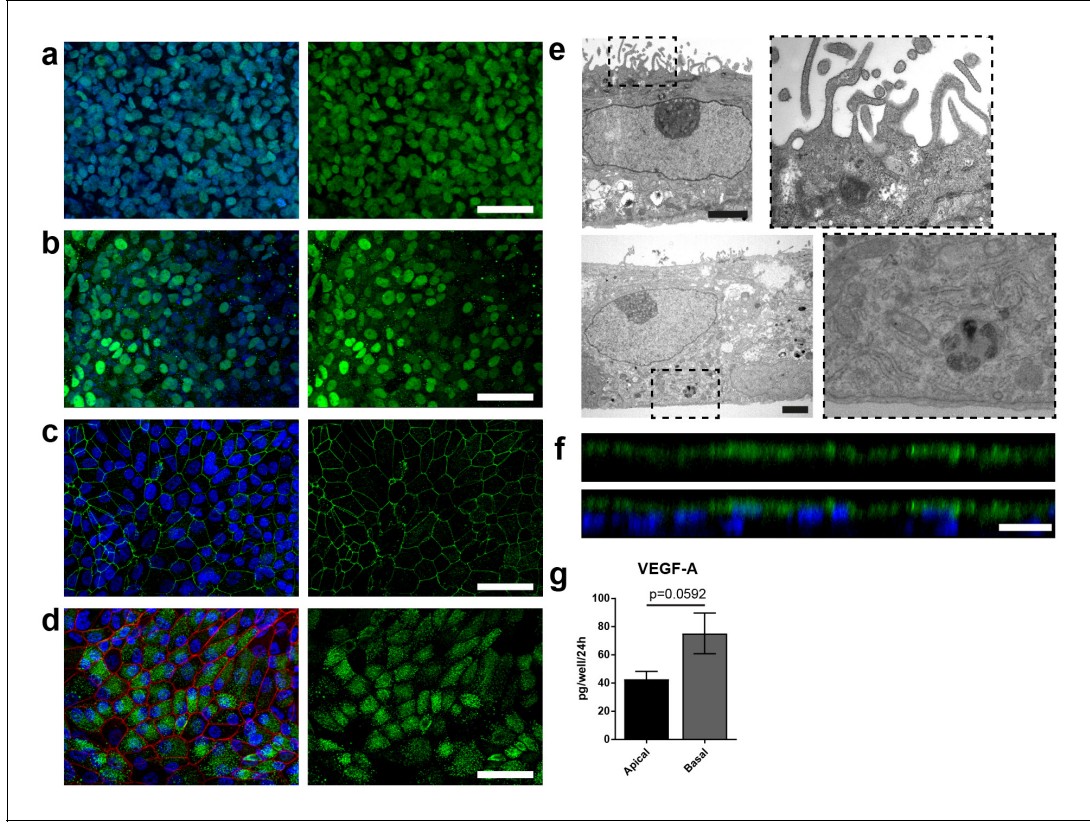

**Figure 4.** Specific marker expression and polarization of retinal pigment epithelial cells in the RoC. (a-d) Evaluation of RPE cells cultured for 14 days in the RoC by immunostaining of relevant RPE markers: a) RPE cells stained for MITF (green), (b) PAX6 (green), (c) ZO-1 (green) and (d) Melanoma gp100 (green), ZO-1 (red). (e) Electron microscopic image of polarized RPE cells. RPE cells display apical microvilli (top row) and a basal membrane (bottom row). (f) Apical microvilli formation is shown using confocal microscopy (orthogonal view of a z-stack) and immunohistochemical staining for ezrin (green). (g) Fluorescent quantification of VEGF-A secretion using ELISA comparing medium collected from a basal and apical channel in a specialized version of the RoC (n = 3 chips). Bars indicate a-d) (left) 50 µm, e) 2 µm, f) 20 µm. Blue: DAPI. Error Bars: S.E.M. p=p value (Two-sided student's t-test).
DOI: https://doi.org/10.7554/eLife.46188.007

The following source data and figure supplements are available for figure 4:

**Source data 1.** Source data for *Figure 4a*.
DOI: https://doi.org/10.7554/eLife.46188.010
**Figure supplement 1.** Characterization of dish and chip cultured human iPSC-derived RPE.
DOI: https://doi.org/10.7554/eLife.46188.008
**Figure supplement 1—source data 1.** Source data for *Figure 4—figure supplement 1a*.
DOI: https://doi.org/10.7554/eLife.46188.009

PMEL17 or *Silver* locus protein) (*Berson et al., 2001*; *Theos et al., 2005*) was highly expressed in chip-cultured RPE (*Figure 4d*) indicating a strong pigmentation. Conclusive evidence for the maturation and proper functionality of RPE is its state of polarization (*Marmorstein, 2001*; *Sonoda et al., 2009*). Electron microscopy analysis revealed not only the strong pigmentation of the RPE but also the presence of apical microvilli as well as a basal membrane already after 7 days of on-chip culture (*Figure 4e*). Further, we observed the polarized expression of ezrin, an apical microvilli marker (*Kivelä et al., 2000*) (*Figure 4f*). Finally, polarized RPE displayed basal secretion of VEGF-A, which could be measured on-chip by using a double-channel chip in which basal and apical medium could be collected separately (*Figure 4g*). The VEGF-A concentration was higher in the basal channel than in the apical (70 vs 40 pg per chip in 24 hr). Taken together, the RPE in the RoC is strongly pigmented, polarized, and expresses respective RPE markers.

## Physiological secretion kinetics into the vasculature-mimicking channels

The vasculature-like perfusion in the media channels enables both, the precisely controllable delivery of defined media and compounds to the tissue as well as the transport of secreted factors away from the tissue, allowing for a time-resolved sampling of the secretion kinetics. In order to analyze and characterize the fluid flow as well as the transport of diluted species in the RoC, we performed computational fluid dynamics simulations: Due to the fluidic resistance of the porous membrane, the convective fluid flow is confined to the media channels (*Figure 5a*). At the same time, nutrients, compounds, and further dissolved molecules are transported to the tissue chamber via diffusion. This rapid process enables a precisely controllable delivery (*Figure 5b*) as well as a controlled washout. In order to verify this, we conducted a proof of concept experiment during which we switched from a colorless liquid to a colored one and observed a complete distribution of the dye within 300 s (flow rate of 20 µl/hr) (*Figure 5—figure supplement 1*). Subsequent injection of a colorless liquid again demonstrated a washout in the same time frame (see *Video 1 and 2*).

To further elucidate the advantages of the vasculature-like perfusion, we injected media supplemented with 10 ng/ml TGF-β1 over 24 hr and subsequently washed the stimulant out again. By sampling the effluent from the media outlets of the RoC, we were able to measure the VEGF-A kinetics before and during the stimulation as well as after the washout (*Figure 5c*). Already after 2 hr, we could observe a 2-fold increase of VEGF-A levels in the effluent medium relative to the baseline level at 0 hr. After this initial peak, the VEGF-A levels decreased over time resting above the baseline level. Finally, after 24 hr a second peak was reached. The subsequent washout of TGF-β1 using normal media led to a steady decrease of VEGF-A levels at the 36 and 48 hr time-point, respectively. In summary, the vasculature-like perfusion in the RoC enabled the controlled delivery and washout of the stimulant TGF-β1 without disturbing the culture conditions as well as time-resolved monitoring of physiological VEGF-A secretion kinetics.

## Enhanced outer segment formation in the RoC

The close proximity and the precisely orchestrated interaction of photoreceptors and the RPE layer is fundamental for vision, ensuring the phagocytosis and processing of shed photoreceptor outer segments (POS) as well as a supply of nutrients and oxygen (*Elman et al., 1976*). The RoC device allows the establishment of a defined interaction site between the segment structures of the RO and RPE cells without impairing neither structure nor viability of the organoid (*Figure 6—figure supplements 1–2*). Live cell imaging in the chip was enabled by transducing RPE cells with an IRBP-GFP viral vector and by marking the surface structures of the organoids with PNA lectin coupled to Alexa Fluor 568 prior to on-chip culture (*Figure 6a*). By measuring the distance between lectin-marked segment tips and GFP-labeled RPE, we found a distance of approximately 5 ± 3.19 µm over different experiments (*Figure 6c*). Subsequently, immunostaining using rhodopsin (rod outer segments) and phalloidin (cytoskeleton of the RPE and the RO including tight junctions) revealed that the segment structures and RPE cells are in close apposition on-chip (*Figure 6b*).

To further study the mechanically delicate interaction site between ROs and RPE in cryosections, we performed immunofluorescence analysis using a specifically tailored chip version (*Figure 6d*). After 7 days of on-chip culture, the close proximity of RPE and RO was preserved and no indication of cell outgrowth or general loss of integrity of the continuous OLM, labeled by the actin-cytoskeleton marker phalloidin, was observed (*Figure 6d*). Further, the hydrogel-filled space between the

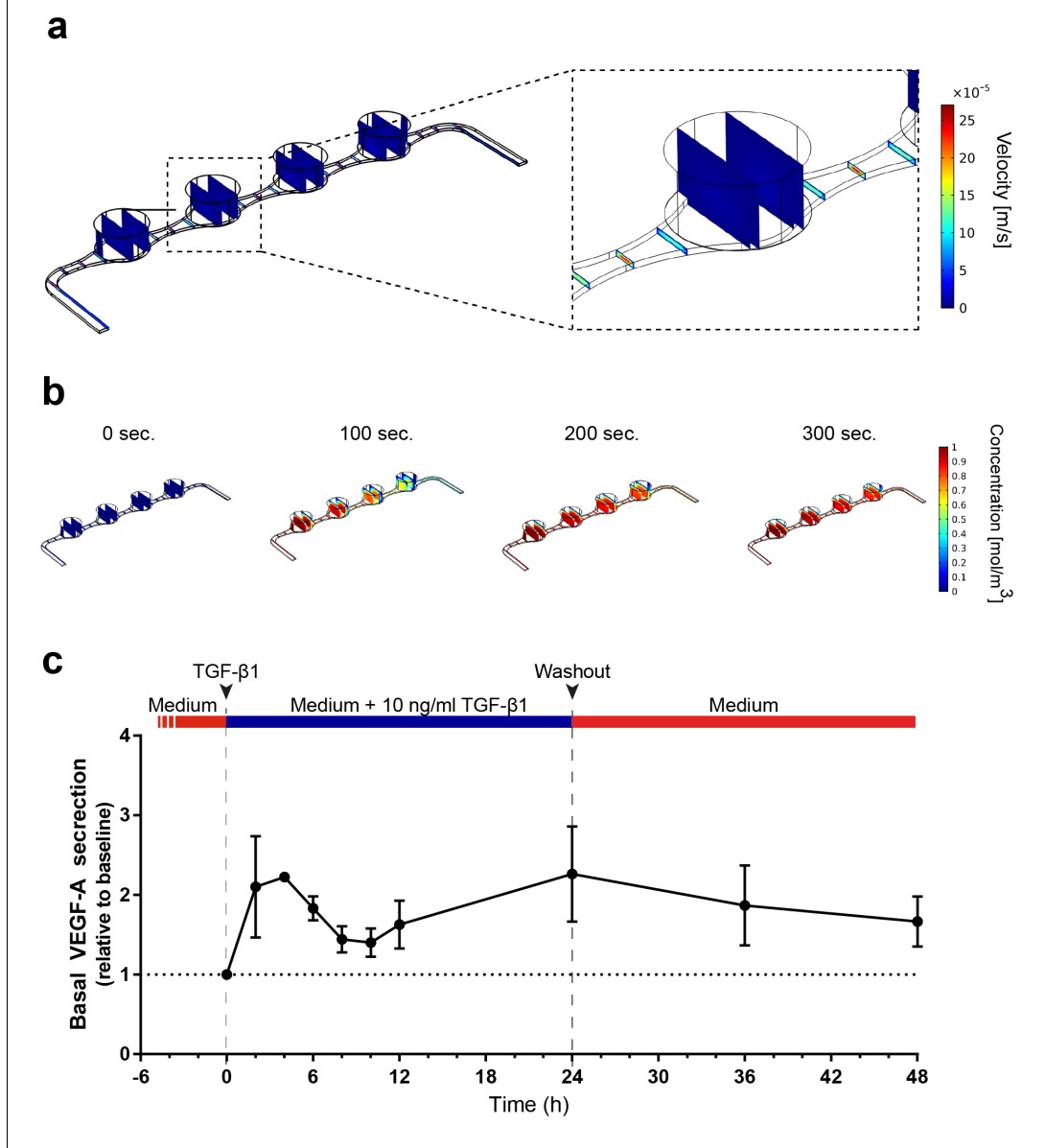

**Figure 5.** Precisely controllable delivery of stimulants and time-resolved secretion kinetics. (a-b) Analysis of the free and porous media flow and the transport of diluted species in the RoC: (a) Convective flow is confined to the vasculature-like media compartments and (b) compounds are delivered rapidly into the tissue chambers via diffusion. (c) Time-resolved monitoring of the secretion of VEGF-A before and after media supplemented with 10 ng/ml TGF-β1 was injected into RoCs (n = 3). After 24 hr, TGF-β1 was washed out using a normal medium. VEGF-A secretion in individual RoCs was normalized to the baseline secretion at 0 hr. Error bars: S.E.M.

DOI: https://doi.org/10.7554/eLife.46188.011
The following source data and figure supplement are available for figure 5:

**Source data 1.** Source data for *Figure 5c*.
DOI: https://doi.org/10.7554/eLife.46188.013
**Figure supplement 1.** Perfusion-enabled precisely controllable injection and washout.
DOI: https://doi.org/10.7554/eLife.46188.012

RPE monolayer and the RO was strongly invaded by rhodopsin and ROM1 (outer segment marker)-positive clusters indicating an increased and cumulated presence of inner and outer segment-like structures (*Figure 6d*). A detailed analysis of the interaction site via electron microscopy confirmed the formation of numerous inner segments with clusters of mitochondria as well as the maintenance

of the OLM (*Figure 6e*). The distance of outer segment tips and RPE microvilli in this exemplary image is around 5 µm (*Figure 6e*), which is in accordance with the data shown above.

Next, we examined whether the formation and preservation of outer segment-like structures are improved in the RoC in comparison to an RO cultured in the chip without RPE and conventional dish cultured ROs (*Figure 6f*). Using electron microscopy, we could find outer segment-like structures in all conditions (*Figure 6f*), displaying distinct disk formation (*Figure 6g*, exemplarily). However, in the RoC system (RoC), the number of outer segment structures was about three times higher than in RO chips without RPE (RoC w/o RPE) and in dish cultured ROs (*Figure 6h*). Interestingly, there was no difference observed in the RoC chip without RPE and dish culture, indicating a positive effect of the RPE on outer segment formation and preservation. In summary, the RoC increases the formation of outer segment-like structures on the RO without disturbing the normal survival and makeup of the organoid structure.

## Modeling key functionalities of the visual cycle

To assess whether the ROs and the RPE can reproduce principle retinal functionality on-chip, we first assessed the ability of the RO photoreceptors to produce an in vivo like calcium flux. Calcium ions ($Ca^{2+}$) are fundamentally important for the function of photoreceptors and involved in many processes ranging from photodetection, transduction and synaptic transfer (reviewed in *Krizaj and Copenhagen, 2002*). To be able to monitor calcium transients and investigate photoreceptor metabolism and functionality, we established on-chip calcium imaging as an easy-to-perform read-out method. By loading RO and RPE on-chip with the calcium dye fura-2-AM, we were able to image calcium dynamics for individual photoreceptors or RPE cell ROIs over an extended timespan (*Figure 7—figure supplement 1*).

Second, we focused on one of the main functions of RPE, the phagocytosis of substances and cell remnants in the form of membrane stacks produced by the photoreceptors. The general capability of the hiPSC-RPE to perform phagocytosis was initially verified using bovine retinal outer segments (*Figure 7—figure supplement 2*). In the RoC platform, RPE cells were labeled by a promoter-driven GFP to mark cell bodies. By labeling the organoids with PNA lectin before the RoC was set up, segment structures were labeled and visualized (as described in *Figure 6a*). Already after 1 day in culture, PNA lectin-positive structures were found within the RPE cell bodies, indicating ongoing digestion of segment particles (*Figure 7a*). The composition of these particles was examined by immunostaining for rhodopsin (*Figure 7b*). This revealed that many of the lectin-positive particles found in the RPE cells were positive for rhodopsin. Next, we examined whether the particles taken up by the RPE are found in the early endosomes, which, in a later step, fuse to phagolysosomes for a full digestion. We labeled RPE cells prior to setting up the RoC with a GFP construct visualizing early endosome complexes (*Figure 7c*, red). After on-chip immunostaining of rhodopsin, specific co-localization of GFP-labeled early endosomes with rhodopsin-positive fragments was detected (*Figure 7c*).

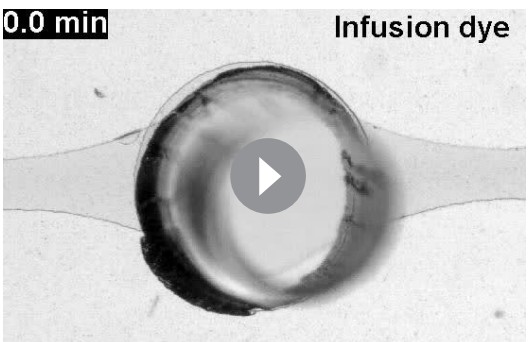

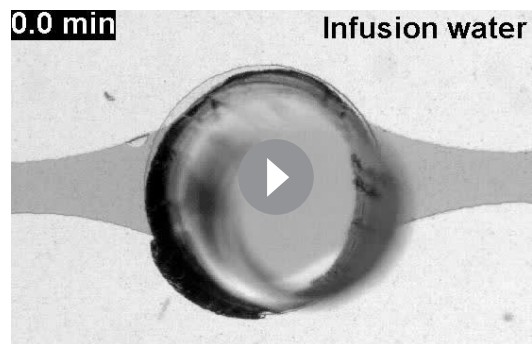

**Video 1.** Bright field microscopy movie depicting the infusion of a colored liquid into a RoC previously perfused with color-less liquid (flow rate 20 µl/hr).
DOI: https://doi.org/10.7554/eLife.46188.014

**Video 2.** Bright field microscopy movie depicting the wash-out of the colored liquid via perfusion with color-less liquid (flow rate 20 µl/hr).
DOI: https://doi.org/10.7554/eLife.46188.015

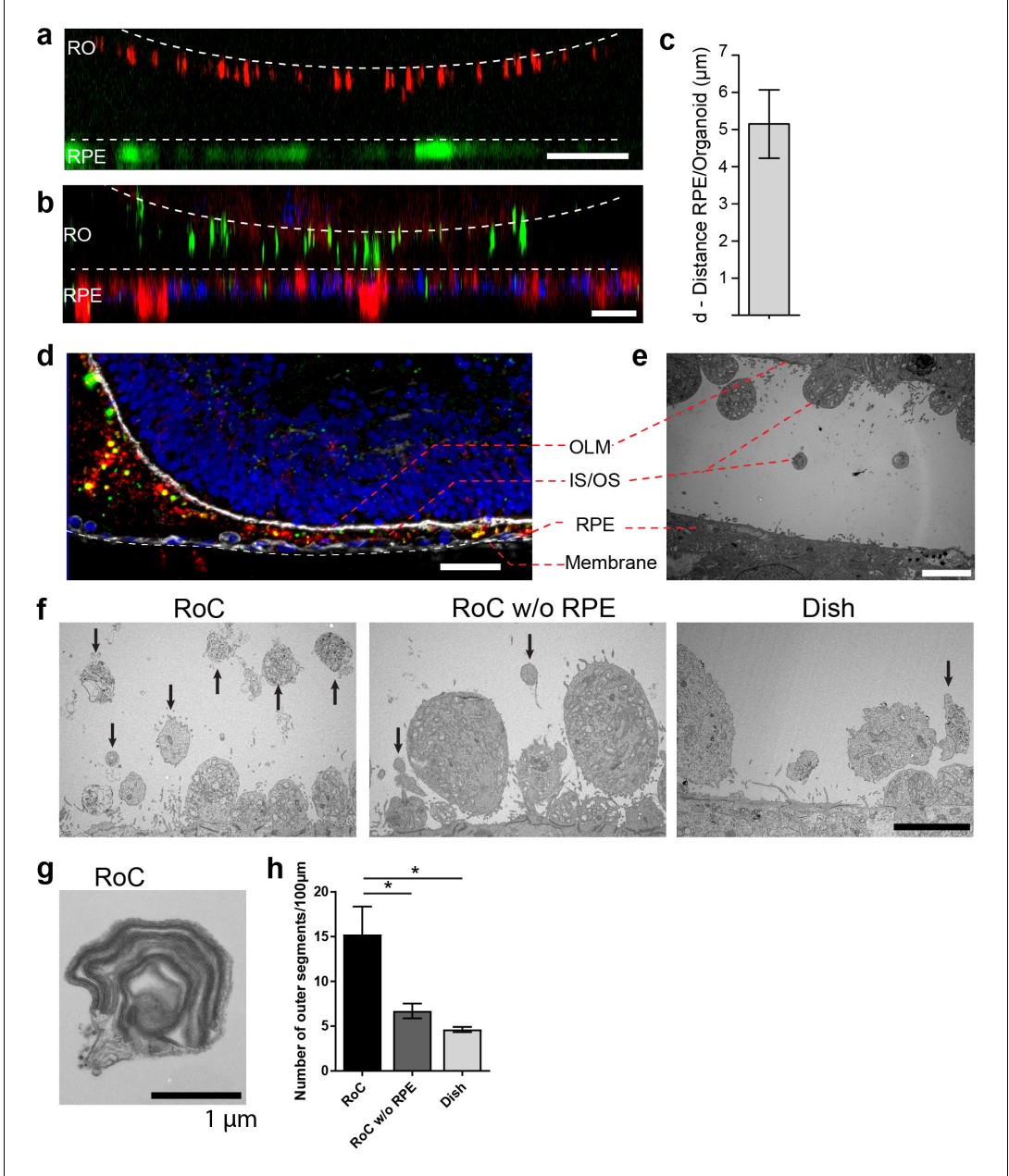

**Figure 6.** RO-RPE interaction enhances the outer segment number at the connection site. (a) For live-cell observation of RoCs, hiPSC-RPE was labeled with a pJG-IRBP-GFP viral vector prior co-culture (green); hiPSC-derived RO-RPCs were labeled with PNA lectin Alexa Fluor 568 (red). RO-RPE interaction site is illustrated as an orthogonal view (b) Orthogonal view of RO (Day 181) and RPE co-cultured for 7 days in the RoC and subsequently stained in situ for rhodopsin (green) and phalloidin (red). c) Distance between GFP-labeled RPE and PNA lectin-stained segment tips in a) was assessed by measurement using orthogonal images (n = 12 chip compartments). (d) Representative cryosection from 7 days co-cultured day 260 ROs and hiPSC-RPE. Sections were immunostained with ROM1 (green; outer segment marker), phalloidin (white; cytoskeleton) and rhodopsin (red; rods). (e) Electron microscopic image of a day 190 organoid facing RPE cultured in the RoC. (f) Representative electron microscopic images of inner and outer segments on the surface of day 181 ROs cultured for 7 days in f1) the RoC, f2) the RoC without RPE and f3) dish-cultured ROs. Black arrows indicate outer segments identified by stack formation. (g) Exemplary high magnification image of an outer segment-like structure containing organized membrane disks found in day 181 ROs cultured for 7 days in the RoC with RPE. (h) Number of segments/100 μm RO circumference comparing RoC, RoC without RPE and dish cultured RO. In the RoC, only the RPE facing side was analyzed (n = 3 RoC, 4 RoC w/o RPE and 3 dish cultured ROs were analyzed). Bars indicate (a-b) 40 μm, (d) 40 μm, (e-f) 5 μm, (g) 1 μm. Blue: DAPI. IS = inner segment, OS = outer segment. Error Bars: S.E.M. *p-value<0.05.
DOI: https://doi.org/10.7554/eLife.46188.016

The following source data and figure supplements are available for figure 6:

**Source data 1.** Source data for *Figure 6c*.

*Figure 6 continued on next page*

*Figure 6 continued*

DOI: https://doi.org/10.7554/eLife.46188.019

**Source data 2.** Source data for *Figure 6h*.

DOI: https://doi.org/10.7554/eLife.46188.020

**Figure supplement 1.** Comparison of dish and chip cultured human iPSC-derived retinal organoids.

DOI: https://doi.org/10.7554/eLife.46188.017

**Figure supplement 1—source data 1.** Source data for *Figure 6—figure supplement 1b*.

DOI: https://doi.org/10.7554/eLife.46188.021

**Figure supplement 2.** Comparison of cell death in RO cultured in the RoC or dish.

DOI: https://doi.org/10.7554/eLife.46188.018

Finally, we visualized the RPE endosomes in the RoC using electron microscopy. We examined day 7 RoCs, identifying indigested outer segment-like structures in the RPE (*Figure 7d*). Here, we found several membrane stack-structures in the RPE below the RO. These structures displayed multi-membrane formation (red arrow) as well as small round membrane structures, both strongly reminiscent of similar features found in outer segment-like structures in ROs (*Figure 7d*, right). Taken together, this strongly indicates functional indigestion of segment structures by the RPE, which is a major prerequisite for a functional visual cycle and therefore a physiological RPE-photoreceptor model.

## Evaluation of drug-induced retinopathy

In order to highlight the RoC's applicability for drug development and toxicology assessment, we exposed the system to the anti-malaria drug chloroquine (CQ) and the antibiotic gentamicin (GM), which both were previously shown to have pathological side effects on the retina (*Elman et al., 1976*; *Ding et al., 2016*; *Yusuf et al., 2017*; *Zemel et al., 1995*; *McDonald et al., 1986*).

After 3 days of on-chip culture, retinal tissue was exposed to two different concentrations of CQ (20 µg/ml and 80 µg/ml) for three additional days. Concentrations were chosen based on previously described effects of CQ on cell viability using the RPE cell line ARPE-19 (*Chen et al., 2011*), and preliminary experiments using hiPSC-RPE for CQ treatment (*Figure 8—figure supplement 1*). Subsequent to the treatment, the RoCs were stained with propidium iodide (PI) to assess cell death (*Figure 8a*). Additionally, they were co-stained with the lysosomal marker protein LAMP2 (*Figure 8c*) since lysosomal dysfunction is involved in the pathophysiology of CQ (*Chen et al., 2011*; *Mahon et al., 2004*; *Rosenthal et al., 1978*). When RoCs were exposed to 20 µg/ml CQ, no significant impact on cell viability (*Figure 8b*) and only a minor increase in LAMP2 signal (*Figure 8c*) were observed. However, at a concentration of 80 µg/ml, cell viability was clearly impacted as shown by a significantly stronger PI staining (*Figure 8a,b*) compared to controls without CQ treatment. Furthermore, after exposure to 80 µg/ml CQ, a strong LAMP2 signal was visible (*Figure 8c*), indicating an enlargement of lysosomes where the drug is accumulating and leading to lysosomal dysfunction. The increase in LAMP2 was not limited to the RPE but was also very pronounced in the RO (*Figure 8c*).

To assess the effect of GM, the antibiotic was added for 6 days to the RoC and to RoC without RPE (*Figure 8d,e*). In RoCs without RPE, an increase in cell death was observed at a GM concentration of 0.5 mg/ml (*Figure 8f*) and even more prominent in RoCs exposed to a 5-fold higher GM concentration (2.5 mg/ml), which was significant in comparison to the controls. In the complete RoC (RO and RPE), similar effects became apparent: The low GM concentrations led to a profound yet not significant increase in the PI signal, whereas the high concentration of 2.5 mg/ml GM led to a significant strong increase (*Figure 8g*). Since the quantified PI signal was a combined signal from cells in RPE and RO, we investigated whether the RO was affected differently by the drug when comparing the conditions with and without RPE (RoC and RoC w/o RPE). For that purpose, we subtracted the PI signal localized in the RPE from the calculated values of the entire RoC in the 0.5 mg/ml treated chips (*Figure 8h*). Interestingly, we found an increase of PI in the condition without RPE, but an unchanged PI signal in the RoC-cultured RO. This is in contrast to the results from the RoC without RPE, where a robust increase was observed when treating the chip with 0.5 mg/ml GM (*Figure 8h*). This could indicate a barrier or even protective function of the RPE, shielding the organoid from the drug and decreasing the toxic effects of the drug on the organoid.

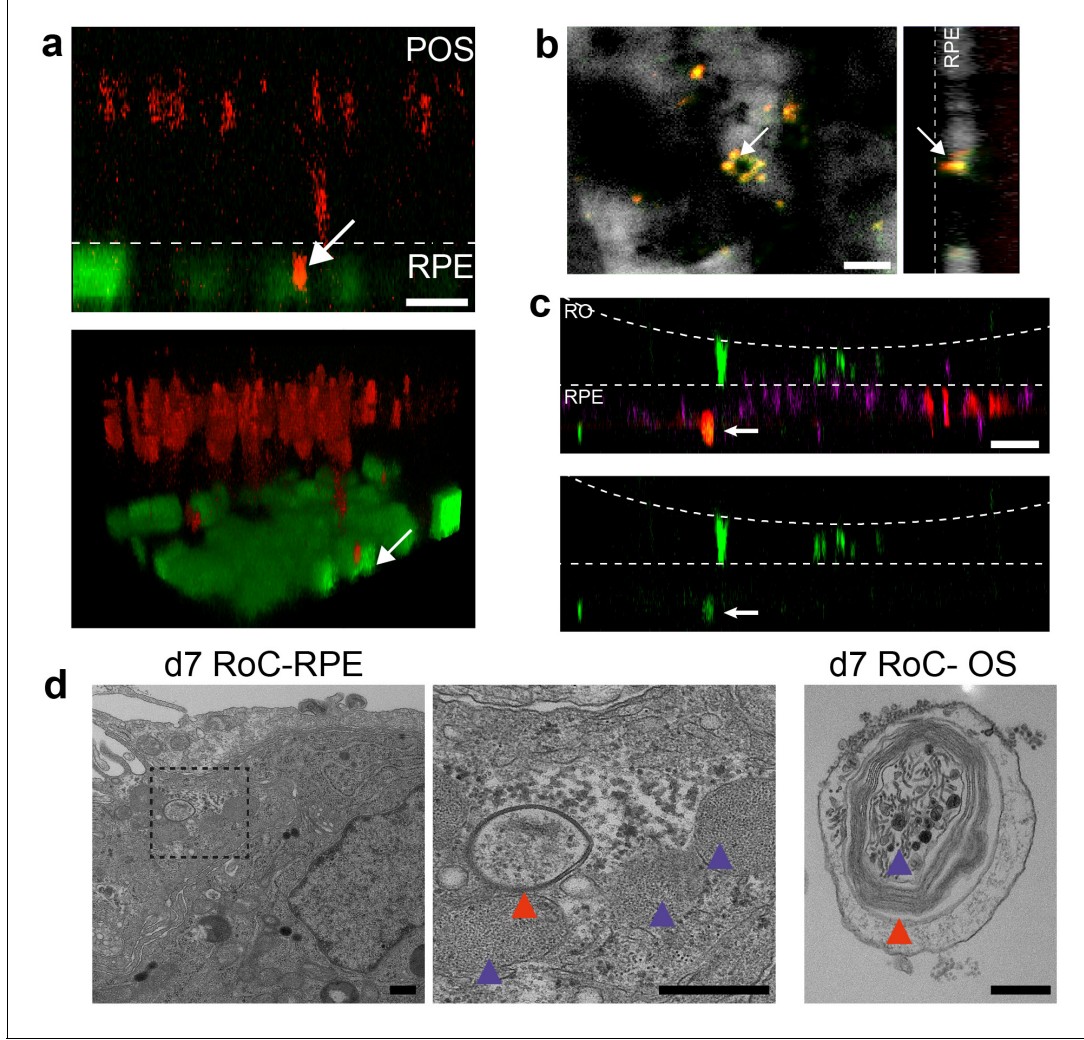

**Figure 7.** Interaction of RO and RPE in the RoC displays phagocytosis of outer segment-like structures. (**a**) Orthogonal view (x-z projection) and 3D reconstruction (bottom) of live-cell-monitored ROs and RPE at day 1 of RoC (RoC) culture. RO (red) and RPE (green) marked as described in *Figure 6a*. Arrow indicates PNA lectin stained photoreceptor segment fragment internalization by RPE cells. (**b**) Samples, as described in a) stained with rhodopsin antibody (red) and PNA lectin (green). RPE (white). Arrows indicate PNA lectin-marked fragments which perfectly co-localize with rhodopsin. The left image shows a top-view; right image an orthogonal y-z projection (**c**) Immunofluorescence imaging of RoC (ROs at day 190 of differentiation). Previously to the chip culture, RPE cells were labeled with an early endosome-GFP construct (red). Chips were thereafter immunostained for rhodopsin (green). (**d**) Electron microscopic images of day 7 RoC. d1) shows RPE situated underneath the RO. d2) magnification of d1) as indicated by the dotted black square. d3) Outer segment-like structure in a day 7 RoC. Red and blue arrows indicate segment-disk structures within the RPE (d2) and the corresponding structures found in an RO outer segment (d3). Scale bars: (**a**) 10 μm, (**b**) 10 μm, (**c**) 50 μm, (**d**) 500 nm Blue: DAPI.

DOI: https://doi.org/10.7554/eLife.46188.022

The following figure supplements are available for figure 7:

**Figure supplement 1.** Phagocytosis assay in dish cultured hiPSC-derived RPE.

DOI: https://doi.org/10.7554/eLife.46188.023

**Figure supplement 2.** Calcium-imaging in the RoC (at 370 nm) with ratiometric calcium indicator dye Fura-2.

DOI: https://doi.org/10.7554/eLife.46188.024

## Discussion

Microphysiological OoC platforms have the potential to revolutionize drug development and may provide new fundamental insights into development and disease. Over the last decade, bioengineering approaches have led to the development of functionally and structurally highly advanced MPSs for a variety of organs and tissues. To study degenerative retinal diseases and investigate retinal

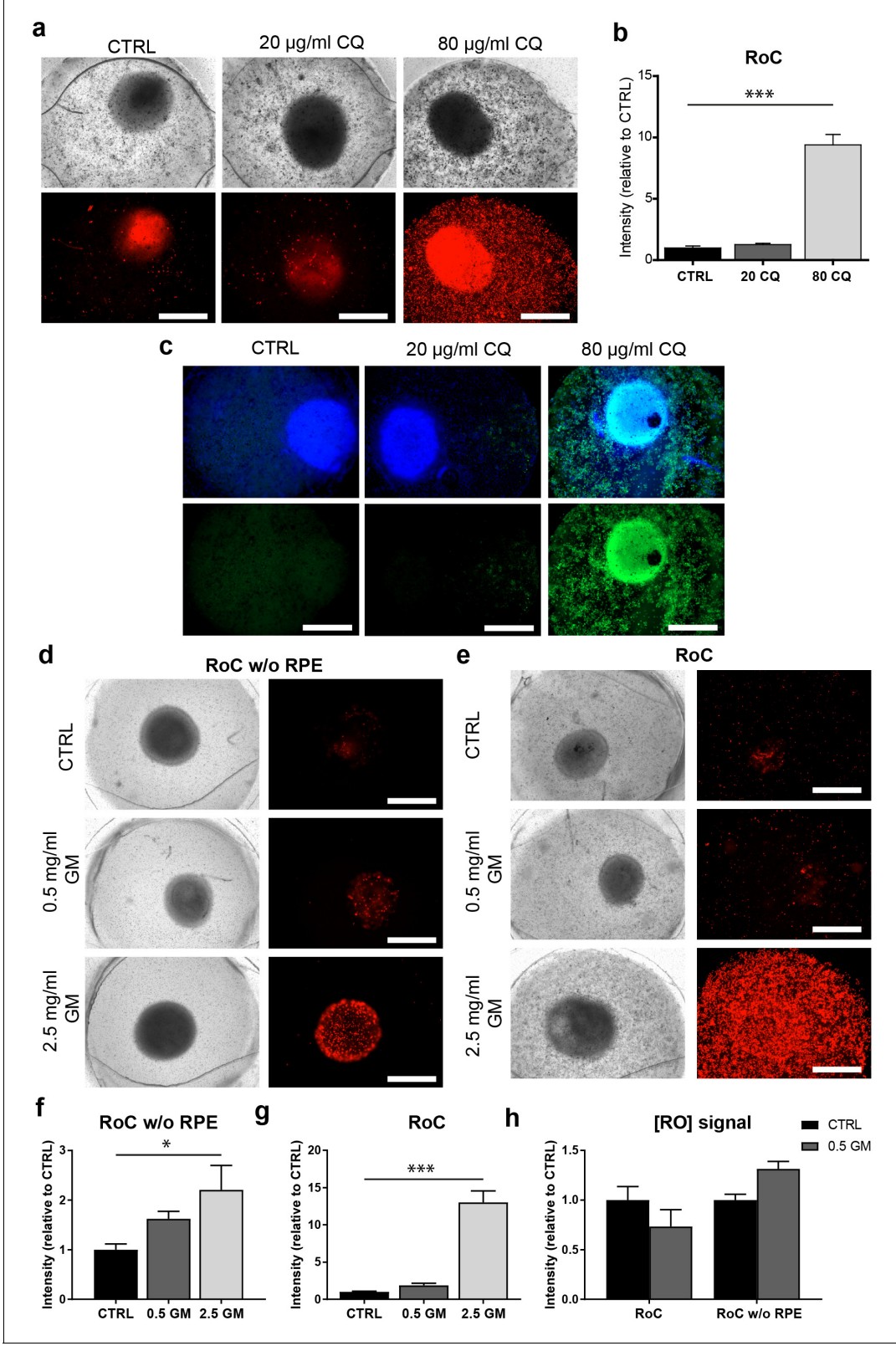

**Figure 8.** The RoC as a pharmacological testing platform. (**a**) Representative bright-field and fluorescence images of day 243–260 RO cultured in the RoC with RPE on day three after treatment with chloroquine (CQ). RoC were either not treated with chloroquine as control (CTRL), treated with 20 μg/ml CQ or treated with 80 μg/ml CQ for 3 days. On day 3, chips were stained with propidium iodide (PI) (red; cell death). (**b**) Quantification of fluorescence intensities of PI staining, relative to controls (n = 6–11 chip compartments in three independent experiments). (**c**) Immunostaining with

*Figure 8 continued on next page*

*Figure 8 continued*

LAMP2 (green, lysosomes) and HOECHST (blue) after 3 days of drug testing of untreated RoCs (CTRL) and RoCs treated with 20 µg/ml CQ or 80 µg/ml CQ. (d) Representative bright-field and fluorescence images of day 202 RO cultured in the RoC without RPE. Cells were treated for 6 days with 0.5 mg/ml gentamicin, 2.5 mg/ml gentamicin or $H_2O$ (CTRL). On day 6, RoCs were stained with propidium iodide (PI) (red, cell death). (e) Representative bright-field and fluorescence images of day 202 RO cultured in the RoC. Cells were treated for six days with 0.5 mg/ml gentamicin, 2.5 mg/ml gentamicin or $H_2O$ (CTRL). On day 6, RoCs were stained with propidium iodide (PI) (red, cell death). (f) Quantification of fluorescence intensities of the PI signal of RO chip compartments without RPE relative to controls (n = 3 chip compartments per conditions). (g) Quantification of fluorescence intensities of the PI signal in the co-culture RoC relative to controls (n = 9 chips compartments per condition). (h) Comparison of the fluorescence intensities of PI staining of RO cultured in the RoC with (left bars) and without RPE (right bars) treated for 6 days with gentamicin (0.5 mg/ml) relative to their individual controls (CTRL). CQ = Chloroquine, GM = Gentamicin. HOECHST (blue; nuclei). Scale bars: 500 µm. Error bars: S.E.M. *p<0.05, ***p<0.001.

DOI: https://doi.org/10.7554/eLife.46188.025

The following source data and figure supplement are available for figure 8:

**Source data 1.** Source data for *Figure 8b*.
DOI: https://doi.org/10.7554/eLife.46188.027
**Source data 2.** Source data for *Figure 8f*.
DOI: https://doi.org/10.7554/eLife.46188.028
**Source data 3.** Source data for *Figure 8g*.
DOI: https://doi.org/10.7554/eLife.46188.029
**Source data 4.** Source data for *Figure 8h*.
DOI: https://doi.org/10.7554/eLife.46188.030
**Figure supplement 1.** Chloroquine applied on dish cultured hiPSC-RPE.
DOI: https://doi.org/10.7554/eLife.46188.026

toxicities, an MPS integrating physiologically relevant retinal tissue is of utmost importance. However, it is extremely challenging if not almost impossible to recapitulate the complex stratified (and interconnected) tissue architecture of the human retina solely using engineering approaches, commonly applied in the field. To create a 3D RoC, we addressed this challenge by combining the biological self-assembly capabilities of ROs with the precisely controllable assembly in microfabricated modules provided by engineering strategies. This combination of interdisciplinary approaches enabled us to successfully create a complex multi-layer structure that includes all cell types and layers present in the neuroretinal ROs, connected to an RPE layer. All integrated cell types were thereby derived from the same hiPSCs. For the first time, we demonstrate a retina model successfully recapitulating the precisely orchestrated interaction between photoreceptors and RPE in vitro. This interaction is one of the key characteristics of the visual cycle, and the RPE is essential for the normal function and survival of photoreceptors, for example via an active phagocytic uptake of photoreceptor outer segments (**POS**) (*Kevany and Palczewski, 2010*). In addition, the microfluidic concept of the RoC adds a further important aspect, the vasculature-like perfusion. The precisely controllable perfusion enables the generation of a physiological transport (both towards and away from the tissue) of nutrients, compounds, and metabolic products, the maintenance of stable, constant conditions over long time-periods (e.g. nutrient and metabolites levels/gradients), as well as the capability to probe the secretome and metabolome in a time-resolved manner. Moreover, it makes the system amenable for the interconnection with further organ-systems enabling the study of for example systemic effects. The applicability of the RoC for compound screening and toxicological studies was demonstrated by i) the successful recapitulation of side-effects of the anti-malaria drug chloroquine and ii) the mimicry of gentamicin-induced retinopathy revealing a protective effect of the RPE barrier. Both the tight barrier function of the RPE layer as well as the melanin-binding of GM (known to be protective in ocular pigmented vs. albino animals (*Zemel et al., 1995*)) could be the source of this protective effect.

In comparison to the conventionally employed retinal model system, the introduced RoC features a variety of advantages and novel potential:

Traditional monolayer cell culture assays have been of limited value in retinal research as they solely include certain cell subtypes, thereby providing a restricted physiological relevance. The more complex ROs solved this issue partially (*Zhong et al., 2014*), but were still limited due to the absence of functional interaction with the RPE. Attempts in recreating the retinal niche in the past have failed to recapitulate the precise RPE-PR arrangement, and thus, did not yield matured

photoreceptors with large formations of membrane disk-containing outer segments. The RoC platform is able to mimic this particular niche and a physiological interaction of RPE and photoreceptor outer segments (POS), by embedding ROs and RPE in the hyaluronic-based hydrogel in specifically tailored microfluidic chambers. This arrangement is much more physiological and controlled than previous approaches employing an unpredictable and unorganized RPE formation during RO generation. This interphotoreceptor matrix in the RoC provides mechanical stability for the small and fragile developing POS, which would otherwise either be damaged or prevented from outgrowth as in conventional suspension cultures. In the RoC, hence, numerous outer segments facing towards the RPE were formed. The flexible and highly controlled tissue assembly paves the way for the modeling of a variety of disease states, for example by combining ROs derived from patients suffering from retinal diseases with RPE derived from the same or healthy donors.

Post-mortem human retinal explants are the onlyhuman models that are currently available and achieve a comparable level of complexity. Those ex-vivo models are, however, greatly limited in terms of supply, with respect to long-term culture, and due to inter-patient variability. Moreover, they are not applicable for studies targeting developmental aspects. The introduced RoC system is entirely based on hiPSCs that are easy to expand and to cryo-conserve. This not only avoids the problem of limited supply but also provides the capability to create a multitude of genetically identical systems and sets the foundation for a scale-up to higher throughput systems, provided an appropriate framework and an automated process landscape is established (*Probst et al., 2018*). The hiPSC technology further paves the way for the generation of disease-specific as well as patient-specific models opening up future applications in disease modeling and personalized medicine. Additionally, hiPSC derived ROs exhibit the ability to stay viable for more than a year in vitro. This is a crucial aspect in terms of answering developmental aspects, studying disease initialization and progression as well as assessing long-term effects or side effects of drugs.

Both current in vitro and ex vivo models share one major limitation, the lack of vascularization or vasculature-like perfusion. This aspect of the microfluidic RoC adds a further dimension of physiological relevance and advantage over the conventional models as described above.

Animal models are so far the only model systems that feature a blood circulation as well as a structural tissue complexity. Yet, besides issues of ethical concerns, results from animal models are often not translatable to humans as none of the small animals used in the field of retina research is able to fully represent the human retinal system. The human genetic background and recapitulation of human retinal tissue structure, hence, provide the potential for the RoC model to overcome those translation hurdles. Animal models, however, still possess structural elements of the visual systems, for example. optical nerve and inner blood-retinal barrier, which cannot be re-created in vitro, so far.

Taken together, the introduced RoC represents a highly advanced in vitro model, which is not hampered by many of the limitations of conventional (in vitro, ex vivo, in vivo) model systems and which can be the first step towards the reduction and replacement of animal models in the field of ophthalmology. Further development will target challenges of innervation (addition of an optical nerve), incorporation of blood-retinal barriers, the integration into multi-organ platforms, as well as the generation of disease-specific systems.

## Summary

The scarce availability of physiologically relevant in vitro models of the human retina and the limited capability of animal models to recapitulate physiological human responses have i) hampered the development of potential new drugs to treat degenerative diseases such as Stargardt disease, age-related macular degeneration or retinitis pigmentosa and ii) prevented the detection of retinal toxicities early in the drug pipeline. By combining hiPSC-ROs and -RPE cells in a microphysiological environment, the introduced human RoC system provides a physiologically relevant model system that recapitulates key functionalities of the human retina, which are impaired in patients suffering from retinal degeneration. Due to a toolbox of in situ and ex-situ analysis options, the platform is extremely versatile and features potential applications for drug development, toxicity screening, disease modeling, and personalized medicine.

# Materials and methods

## Key resources table

| Reagent type (species) or resource | Designation | Source or reference | Identifiers | Additional information |
|---|---|---|---|---|
| Antibody | Donkey anti-Mouse IgG Secondary Antibody Alexa Fluor 488, polyclonal | ThermoFisher Scientific | Cat.# R37114 RRID:AB_2556542 | IHC(1:1000) |
| Antibody | Donkey anti-Mouse IgG Secondary Antibody Alexa Fluor 568, polyclonal | ThermoFisher Scientific | Cat.# A10037 RRID:AB_2534013 | IHC(1:1000) |
| Antibody | Donkey anti-Mouse IgG Secondary Antibody Alexa Fluor 647, polyclonal | ThermoFisher Scientific | Cat.# A-31571 RRID:AB_162542 | IHC(1:1000) |
| Antibody | Donkey anti-Goat IgG Secondary Antibody Alexa Fluor 488, polyclonal | ThermoFisher Scientific | Cat.# A-11055 RRID:AB_2534102 | IHC(1:1000) |
| Antibody | Donkey anti-Goat IgG Secondary Antibody Alexa Fluor 568, polyclonal | ThermoFisher Scientific | Cat.# A-11057 RRID:AB_142581 | IHC(1:1000) |
| Antibody | Donkey anti-Goat IgG Secondary Antibody Alexa Fluor 647, polyclonal | ThermoFisher Scientific | Cat.# A-21447 RRID:AB_141844 | IHC(1:1000) |
| Antibody | Donkey anti-Rabbit IgG Secondary Antibody Alexa Fluor 488, polyclonal | ThermoFisher Scientific | Cat.# R37118 RRID:AB_2556546 | IHC(1:1000) |
| Antibody | Donkey anti-Rabbit IgG Secondary Antibody Alexa Fluor 568, polyclonal | ThermoFisher Scientific | Cat.# A10042 RRID:AB_2534017 | IHC(1:1000) |
| Antibody | Donkey anti-Rabbit IgG Secondary Antibody Alexa Fluor 647, polyclonal | ThermoFisher Scientific | Cat.# A-31573 RRID:AB_2536183 | IHC(1:1000) |
| Antibody | Mouse anti-AP2$\alpha$, monoclonal | Santa Cruz Biotechnology | Cat.# sc-12726 RRID:AB_667767 | IHC(1:100) |
| Antibody | Goat anti-Arrestin 3 (Cone Arrestin), polyclonal | Santa Cruz Biotechnology | Cat.# sc-54355 RRID:AB_2060084 | IHC(1:50) |
| Antibody | Goat anti-Brn-3b, polyclonal | Santa Cruz Biotechnology | Cat.# sc-31989 RRID:AB_2167523 | IHC(1:50) |
| Antibody | Goat anti-Chx10, polyclonal | Santa Cruz Biotechnology | Cat.# sc-21690 RRID:AB_2216006 | IHC(1:200) |
| Antibody | Mouse anti-CRALBP, monoclonal | Abcam | Cat.# ab15051 RRID:AB_2269474 | IHC(1:250) |
| Antibody | Mouse anti-EEA1, monoclonal | ThermoFisher Scientific | Cat.# 14-9114-80 RRID:AB_2572928 | IHC(1:500) |
| Antibody | Rabbit anti-Ezrin, polyclonal | Cell Signaling | Cat.# 3145S RRID:AB_2100309 | IHC(1:200) |
| Antibody | Mouse anti-LAMP2, monoclonal | Santa Cruz Biotechnology | Cat.# sc18822 RRID:AB_626858 | IHC(1:50) |
| Antibody | Mouse anti-Melanoma-gp100, monoclonal | Abcam | Cat.# ab787 RRID:AB_306146 | IHC(1:100) |
| Antibody | Mouse anti-MITF, monoclonal | Exalpha Biologicals | Cat.# X1405M | IHC(1:500) |
| Antibody | Rabbit anti-Pax-6, polyclonal | Covance | Cat.# PRB-278P-100 RRID:AB_291612 | IHC(1:100) |
| Antibody | Rabbit anti-PKC$\alpha$, polyclonal | Santa Cruz Biotechnology | Cat.# sc-208 RRID:AB_2168668 | IHC(1:500) |

*Continued on next page*

*Continued*

| Reagent type (species) or resource | Designation | Source or reference | Identifiers | Additional information |
|---|---|---|---|---|
| Antibody | Mouse anti-Rhodopsin, monoclonal | Santa Cruz Biotechnology | Cat.# sc-57432 RRID:AB_785511 | IHC(1:200) |
| Antibody | Rabbit anti-ROM1, polyclonal | Proteintech | Cat.# 21984–1-AP | IHC(1:200) |
| Antibody | Mouse anti-RPE65, monoclonal | Abcam | Cat.# ab78036 RRID:AB_1566691 | IHC(1:100) |
| Antibody | Rabbit anti-Opsin, blue, polyclonal | Merck Millipore | Cat.# ab5407 RRID:AB_177457 | IHC(1:200) |
| Antibody | Rabbit ani-ZO-1, polyclonal | ThermoFisher Scientific | Cat.# 61–7300 RRID:AB_138452 | IHC(1:100) |
| Commercial assay or kit | TUNEL Assay (Click-iT TUNEL Alexa Fluor 488 Imaging Assay) | ThermoFisher Scientific | Cat.# C10245 | |
| Commercial assay or kit | CellLight Early Endosomes-GFP, BacMam 2.0 | ThermoFisher Scientific | Cat.# C10586 | |
| Commercial assay or kit | VEGF-A Human ELISA Kit | ThermoFisher Scientific | Cat. # BMS277-2 | |
| Chemical compound, drug | Alexa Fluor 647 Phalloidin | ThermoFisher Scientific | Cat.# A12379 | 1:500 |
| Chemical compound, drug | PNA lectin-Alexa Fluor 568 | ThermoFisher Scientific | Cat.# L32458 | 20 µg/ml |
| Chemical compound, drug | PNA lectin-Alexa Fluor 647 | ThermoFisher Scientific | Cat.# L32460 | 20 µg/ml |
| Chemical compound, drug | Recombinant Human TGF-β1 | Peprotech | Cat.# 100–21 | |
| Chemical compound, drug | HOECHST 33342 | ThermoFisher Scientific | Cat.#H3570 | 1:2000 |
| Chemical compound, drug | Chloroquine | Sigma-Aldrich | Cat.#C6628 | |
| Chemical compound, drug | Gentamicin | Sigma-Aldrich | Cat.#G1397 | |
| Recombinat DNA reagent | pJG-IRPB-eGFP | Department of Biological Structure, University of Washington (https://faculty.washington.edu/tomreh/) | | |
| Software, algorithm | COMSOL Multiphysics | COMSOL Multiphysics | RRID:SCR_014767 | |
| Software, algorithm | Prism | GraphPad | RRID:SCR_002798 | |
| Software, algorithm | FIJI | | RRID:SCR_002285 | |

## Fabrication of retina MPS

The RoC consists of two Polydimethylsiloxane (PDMS) layers and a porous Polyethylene terephthalate (PET) membrane in between, bonded to a thin glass slide (170 µm). First, PDMS master molds were fabricated. For the media channel mold, SU8-50 photoresist (MicroChem, USA) was spin-coated onto a previously cleaned 4″ silicon wafer to obtain a height of 100 µm. To create the desired structure, the substrate was exposed to 350 mJ/cm$^2$ of UV light, followed by development in SU-8 developer (Microresist Technology GmbH, Germany) for 6 min. Finally, the wafer was rinsed with isopropanol and blow-dried using nitrogen. The second wafer for RO and RPE culture was fabricated in two steps. Initially, a base layer of 25 µm for the membrane insert was fabricated by spin-coating a first layer of photoresist SU8. The exposure to UV light, in this case, was 200 mJ/cm$^2$. Subsequently,

the wafer was developed in SU-8 developer for 4 min, rinsed in isopropanol and blow-dried with nitrogen. Next, the wafer was coated with a second layer of SU8-3025 to fabricate the tissue channels with a height of 40 µm. The wafer was exposed to UV light at 250 mJ/cm$^2$ for 10 s and developed for 4 min. Afterward, both master molds were silanized with chlorotrimethylsilane (Sigma-Aldrich, Germany). Subsequently, Sylgard 184 PDMS (Dow Corning, USA) was mixed at a 10:1 ratio of prepolymer to curing agent and molded by using the wafers as a negative master mold. The layer for the media supply was made by exclusion molding followed by curing overnight at 60°C. The RO/RPE culture layer was fabricated by pouring 25 g of the PDMS mixture onto the master mold and curing it overnight at 60°C. Next, the PDMS slabs were peeled off the wafers and the media-supply layers were bonded to a glass slide previously cleaned by a 30 s exposure to oxygen plasma at 50 Watts. Inlets and outlets were punched using a biopsy puncher with a diameter of 0.75 mm. To culture the cells and organoids, four chambers were punched out of the PDMS with a biopsy puncher of 2 mm diameter. Semipermeable membranes with a diameter of 20 mm, made from PET (Sabeu GmbH, Germany) with a pore diameter of 3 µm and a thickness of 10–20 µm, were functionalized using bis-[3-trimethoxysilylpropyl]amine (Sigma-Aldrich, Germany). Before assembly, both PDMS layers were cleaned with isopropanol and Scotch tape to remove dust particles. Afterwards, both layers were treated with oxygen plasma at 50 W for 30 s. Then, the membrane was placed into the inlay of the RO/RPE culture layer. Finally, both layers were aligned to each other using a stereo microscope and baked overnight at 60°C to stabilize bonding.

## Cell culture

### iPSC Culture

All hiPSC cell lines were derived from healthy donors as previously described (*Linta et al., 2012*) and tested for stem cell markers and germ-layer differentiation potential. HiPSCs were cultured on Matrigel (hESC-qualified, BD Biosciences, USA)-coated plates with FTDA medium (*Frank et al., 2012*). Cells were passaged every 6–7 days using Dispase (Stemcell Technologies, Canada). Differentiated colonies were removed manually by scraping. All procedures were in accordance with the Helsinki convention and approved by the Ethical Committee of the Eberhard Karls University Tübingen (Nr. 678/2017BO2). Control persons gave their written consent.

### Retinal organoid culture

HiPSC-derived RO were differentiated based on a protocol by *Zhong et al. (2014)* with some modifications. Briefly, for embryoid body (EB) formation, $2.88 \times 10^6$ hiPSCs were detached on day 0 using TrypLE (ThermoFisher Scientific, USA) and dissociated to single cells. Cells were then mixed with PeproGrow (Peprotech, USA) medium, 10 µM Y-27632 (ROCK-inhibitor, Ascent Scientific, USA) and 10 µM blebbistatin (Sigma-Aldrich, USA) and distributed to 96 untreated v-shaped 96-wells (Sarstedt, Germany). For re-aggregation, the plate was centrifuged at 400 g for 4 min. On day 1, 80% of the medium was removed and replaced with N2 medium (DMEM/F12 (1:1)+Glutamax supplement (ThermoFisher Scientific, USA), 24 nM sodium selenite (Sigma-Aldrich, USA), 16 nM progesterone (Sigma-Aldrich, USA), 80 µg/ml human holotransferrin (Serologicals, USA), 20 µg/ml human recombinant insulin (Sigma-Aldrich), 88 µM putrescin (Sigma-Aldrich, USA), 1x minimum essential media-non essential amino acids (NEAA, ThermoFisher Scientific, USA), 1x antibiotics-antimycotics (AA, ThermoFisher Scientific, USA)). Medium was changed again on day 4. On day 7, EBs were plated on Growth-Factor-Reduced Matrigel (BD Biosciences, USA)-coated six well plates at a density of 32 EBs/well and medium was changed daily. On day 16, medium was switched to a B27-based Retinal differentiation medium (BRDM) (DMEM/F12 (3:1) with 2% B27 (w/o vitamin A, ThermoFisher Scientific, USA), 1x NEAA and 1x AA). On day 24, eye fields were detached using 10 µl tips and collected in 10 cm bacterial petri dishes (Greiner Bio One, Germany) with BRDM, adding 10 µM ROCK-Inhibitor Y-27632 for one day. After completed formation, ROs were selected and if necessary detached from non-retinal spheres using microscissors. From day 40 onwards, ROs in BRDM were supplemented with 10% fetal bovine serum (FBS, Thermo Fisher Scientific, USA) and 100 µM taurine (Sigma-Aldrich, USA). From day 70–100, BRDM with FBS and taurine was further supplemented with 1 µM retinoic acid (Sigma-Aldrich, USA), which was reduced to 0.5 µM during days 100–190 and removed afterwards.

### Differentiation of retinal pigment epithelial cells

RPE cells were derived as a product from RO differentiation following (slightly adapted) procedures of *Zhong et al. (2014)* and *Ohlemacher et al. (2015)*. For this purpose, pigmented areas or spheres were removed from ROs using microscissors under an inverted microscope. The pigmented areas were collected in 1.5 ml Eppendorf tubes (Eppendorf, Germany) and washed once with Dulbecco's phosphate-buffered saline (PBS, no calcium, no magnesium, Thermo Fisher Scientific, USA). To dissociate the RPE into single cells for adhesion culture, the pigmented spheres were treated with Accumax (Sigma-Aldrich, USA) for 90 min at 37°C and 5% $CO_2$ and resuspended every 30 min using a 100 µl pipette. The reaction was stopped using BRDM with 10% FBS followed by centrifugation at 1500 rpm for 2 min. The derived single RPE-cells were plated on 6-well plates or coverslips in 24-well plates, treated with a 0.01% Poly-L-Ornithine Solution (Sigma-Aldrich, USA) for 30 min at room temperature and 20 µg/ml Laminin (Roche, Switzerland) for 4 hr at 37°C and 5% $CO_2$. For the plating of the cells, BRDM was supplemented with 20 µg/ml EGF (Cell Guidance Systems, United Kingdom), 20 µg/ml FGF2 (Cell Guidance Systems, United Kingdom), 2 µg/ml heparin (Sigma-Aldrich, USA), and 10 µM Y-27632 (ROCK-inhibitor, Ascent Scientific, USA) (*Ohlemacher et al., 2015*; *Croze et al., 2014*). In addition, for the first 24 hr, 10% FBS (Thermo Fisher Scientific, USA) was added to achieve adherence of the cells. When cells had reached confluence, medium was switched to BRDM without supplementation.

### Transduction of RPE cells

To generate green fluorescent iPSC-RPE lines, adherent RPE cultures were incubated with lentiviral particles generated from pJG-IRPB-eGFP plasmids (*Lamba et al., 2010*) (Gift from Deepak Lamba, Thomas Reh) in BRDM + 10% FBS for one day, washed three times with PBS and further cultivated in BRDM.

### RoC culture

Individual systems were sterilized via oxygen plasma treatment for 3 min at 50 Watts and placed into PBS-filled 50 ml tubes to displace the air in the channels. Before seeding hiPSC-RPE cells into the MPS, each system was removed from the tube, carefully dried with a paper towel and placed into a 10 cm dish. Each well was coated for 2 hr with 50 µg/ml Laminin in DMEM/F12 at 37°C and 5% $CO_2$. RPE cells were detached and dissociated using Accumax at 37°C and 5% $CO_2$ for 10–40 min, depending on the adherence and passage of the cells. To remove cell agglomerates, a 70 µm cell strainer was used. As a next step, each well was seeded with RPE at a density of 27 000 cells in a volume of 4.5 µl BRDM supplemented with 10% FBS. RoC were incubated for at least 2 hr at 37°C and 5% $CO_2$ to allow RPE cells to adhere to the semipermeable membrane. The medium was changed every day for 1–3 days prior ROs were loaded into the RoCs. ROs were placed onto the RPE covered membrane. Hyaluronic acid-based hydrogel HyStem-C (ESI Bio, USA) was prepared according to the manual and added to the well by pipetting. During culture, the chambers were covered by a sterile adhesive tape (optical adhesive covers, Thermo Fischer Scientific, USA) to avoid evaporation. BRDM supplemented with 100 µM taurine and 10% FBS was supplied at a constant flow rate of 20 µl/h by syringe pump.

### Drug treatment

RPE and ROs were seeded into the RoC as described above. Subsequently, ROs and RPE were either treated for three days with 20 and 80 µg/ml chloroquine (Sigma-Aldrich, USA) in BRDM using a syringe pump at a flow rate of 30 µl/h or with 0.5 mg/ml and 2 mg/ml Gentamicin (Sigma-Aldrich, USA) for 6 days. For every treatment, control RoCs were also used, without addition of equal amounts of the solvent ($H_2O$). After 3 days of treatment, cells in the RoCs were stained using HOECHST (Thermo Fischer Scientific, USA) and 3 µM propidium iodide (PI, Sigma Aldrich, USA) to assess cell death. RoCs were washed twice with PBS using a syringe and fixed with 4% PFA for immunohistochemical staining of LAMP2.

### Phagocytosis assay using bovine ROS

Bovine rod outer segments were isolated as previously described (*Vogt et al., 2013*). For the phagocytosis assay out-of-the-chip, hiPSC-RPE was plated on cover slips after coating with 0.01% Poly-L-Ornithine Solution and Laminin as described above. For the phagocytosis assay in the RPE-

chip, RPE was loaded as described above into the Laminin-coated wells. On the next day, hiPSC-RPE on coverslips or in the chip were incubated with bovine photoreceptor outer segments (POS) at a density of 10 POS/RPE in BRDM for 2 hr at 37°C, then washed with PBS 3x and cultivated for additional 2 hr in BRDM and then fixed with 4% paraformaldehyde (Carl Roth, Germany) and 10% sucrose (Carl Roth, Germany) in PBS for 20 min at room temperature for immunohistochemistry.

## VEGF-A secretion assays

Specialized double-channel RoCs were generated for apical and basal secretion measurement. These chips were identical with the previously described setup, except that an additional channel was included (apical channel) connecting the compartments, with an additional in- and outlet that allows media flow above the RPE layer, in addition to the media flow below. These double-channel chips were loaded with RPE cells as described and then cultivated for 14 days using a syringe pump. Effluent from the upper and lower-channel outlet were collected after 24 hr. The apical and basal media were analyzed from three different chips and initial volumina were noted for calculation of the total substance quantity per chip on the apical or basal side.

Stimulation of VEGF-A secretion was measured in regular RoC platforms. To acquire comparable samples, the effluent was collected over 2 hr resulting in 100 μl volumes (flow rate of 50 μl/h). The effluent was collected once before TGF-β1 exposure to measure the baseline secretion of VEGF-A. Subsequently, the RoCs were perfused with medium containing 10 ng/ml TGF-β1 (Peprotech, USA). Samples were collected every 2 hr for 12 hr and once after 24 hr. After 24 hr, TGF-β1 was removed from the medium and samples were collected at the 36 and 48 hr time-points, 12 and 24 hr after starting the washout, respectively.

The collected samples were immediately frozen at −20°C. The concentration of VEGF-A was measured after defrosting the samples using the VEGF-A Human ELISA Kit (Thermo Fisher Scientific, USA). The assay was performed according to the manufacturer's protocol and absorbance was measured at 450 nm.

## Live cell labeling of hiPSC retinal organoids

For live cell labeling of RO photoreceptor segments, ROs were incubated in a reaction tube for 30 min in BRDM containing 20 μg/ml PNA lectin-Alexa Fluor 568 (Thermo Fisher Scientific, USA) or PNA lectin-Alexa Fluor 647 (Thermo Fisher Scientific, USA) followed by washing with medium four times, prior to the transfer into the RoC.

## Live cell endocytosis and phagocytosis assay

For live cell endocytosis experiments, RPE cells were infected overnight with 10 particles/cell of Cell-Light Early-Endosomes GFP (BacMam 2.0, Thermo Fisher Scientific, USA) prior to the seeding of RPE into the RoC. Endosome labeling could be detected for >5 days.

## Production of agarose RoCs and cryoembedding

Agarose RoCs were produced from an in-house fabricated mold using 4% Agarose/BRDM + 10% FBS containing four separate compartments and a semipermeable membrane (as described in the MPS section) at the bottom of each well. RPE and ROs were loaded into the agarose RoCs as already described. For fixation, agarose RoCs were fixed in 4% paraformaldehyde in 0.1 M phosphate buffered saline (pH 7.4) (Polysciences, Warrington Pa., USA) for 2 hr. ROs from classic dish culture were washed with PBS and fixed with 4% paraformaldehyde and 10% sucrose in PBS for 20 min at room temperature, then kept in PBS at 4°C.

After rinsing in PBS, agarose RoCs or RO were cryoprotected in graded sucrose/PBS (10% for 30 min, 20% for 1 hr, 30% overnight), embedded in cryomatrix (Tissue-Tek O.C.T. Compound, Sakura, Netherlands) and frozen in liquid nitrogen. Cryosections (14 μm) were cut on a Leica CM 3050 s Cryocut, mounted on Superfrost glass slides, and stored at −20°C.

## Transmission electron microscopy

For transmission electron microscopy, agarose RoCs with ROs and RPE were fixed in the chambers with Karnovsky buffer (2.5% glutaraldehyde, 2% paraformaldehyde, 0.1 M sodium cacodylate buffer, pH 7.4) (Electron Microscopy Sciences, Germany) for 12 hr at 4°C. After fixation, the samples were

rinsed three times in 0.1 M sodium cacodylate buffer (pH 7.4, Electron Microscopy Sciences, Germany) for a total of 30 min, and postfixed in 1% OsO$_4$ (Electron Microscopy Sciences, Germany) for 1.5 hr at room temperature. After three additional washes in cacodylate buffer and dehydration in 50% ethanol, tissues were counterstained with 6% uranyl acetate dissolved in 70% ethanol (Serva, Germany) followed by graded ethanol concentrations of ethanol (80% and 96% for 15 min each, 100% for two times 10 min, acetone 100%, 15 min). The dehydrated samples were incubated in a 2:1 and 1:1 mixture of acetone and Epon resin (Serva, Germany) for 1 hr each, on a shaker. Finally, organoids were infiltrated with pure Epon and polymerized by overnight incubation at 60°C. The next day, ROs and RPE were punched out of the chambers. Upon punches containing RPE-filter and ROs were embedded in fresh resin in flat molds (Science Services, Germany) and cured 12 hr at 60°C followed by 2 hr at 90°C.

Ultrathin sections (50 nm) were cut on a Reichert Ultracut S (Leica, Germany), collected on copper grids and counterstained with Reynolds lead citrate. Sections were analyzed with a Zeiss EM 900 transmission electron microscope (Zeiss, Germany) equipped with a 2k × 2 k CCD camera.

Images were used for quantification of outer segment density using an image analysis software (iTEM, Olympus Soft Imaging Solutions, Germany). To calculate the ratio of outer segments per μm organoid surface, a line was drawn and measured along the outer limiting membrane of the organoid and outer segment structures visible along the line were counted.

## Immunohistochemistry

For in situ chip staining, whole-mount staining was performed using a blocking solution of 5% or 10% normal donkey serum (Millipore, USA) with 0.2% triton-X (Carl Roth, Karlsruhe, Germany) for permeabilization, twice for 1 hr. Primary antibodies were added to the blocking solution for 1 or 2 days at 4°C, then secondary antibodies were added in blocking solution overnight at 4°C. Next, RoC were counterstained with HOECHST 33342 for 10 min at room temperature (1:2000, Thermo Fisher Scientific, USA). Washing steps to remove residual antibodies were performed with PBS, three times for 2 hr at room temperature after incubation of primary and secondary antibodies, as well as after HOECHST staining.

Cryosections from agarose-chips and ROs were rehydrated in PBS for 15 min and incubated in a blocking solution of 10% normal donkey serum in PBS with 0.2% triton-X for 1 hr. Wholemount ROs were incubated in a blocking solution of 10% normal donkey serum in PBS with 0.2% triton-X for 1 hr. Primary antibodies were diluted in blocking solution and incubated overnight at 4°C. Secondary antibodies were diluted in 1:1 blocking solution:PBS and incubated for 2 hr at room temperature. Mounting was performed with ProLong Gold Antifade Reagent with DAPI (Thermo Fisher Scientific, USA). Washing steps to remove residual antibodies were performed with PBS, three times for 3 min at room temperature after primary and secondary antibodies.

Cells grown on glass coverslip were washed with PBS and fixed with 4% paraformaldehyde and 10% sucrose in PBS for 20 min at room temperature, then kept in PBS at 4°C. For blocking and permeabilization, cover slips were incubated with 5% normal donkey serum and 0.2% triton-X for 1 hr. Primary antibodies were diluted in blocking solution and incubated overnight at 4°C. Secondary antibodies were diluted in blocking solution and incubated for 2 hr at room temperature. Mounting was performed with ProLong Gold Antifade Reagent with DAPI (Thermo Fisher Scientific, USA). Washing steps to remove residual antibodies were performed with PBS, three times for 5 min at room temperature after primary and secondary antibodies.

For LAMP-2 stainings, 0.5% saponin (Millipore, USA) was used instead of Triton-X and washing steps were performed using 0.1% saponin in PBS instead of using only PBS.

Antibodies used were:
Secondary Antibodies:

- Donkey anti-Mouse Alexa Fluor 488 (1:1000, R37114, Thermo Fisher Scientific, USA)
- Donkey anti-Mouse Alexa Fluor 568 (1:1000, A10037, Thermo Fisher Scientific, USA)
- Donkey anti-Mouse Alexa Fluor 647 (1:1000, A-31571, Thermo Fisher Scientific, USA)
- Donkey anti-Goat Alexa Fluor 488 (1:1000, A-11055, Thermo Fisher Scientific, USA)
- Donkey anti-Goat Alexa Fluor 568 (1:1000, A-11057, Thermo Fisher Scientific, USA)
- Donkey anti-Goat Alexa Fluor 647 (1:1000, A-21447, Thermo Fisher Scientific, USA)
- Donkey anti-Rabbit IgG (H + L) Alexa Fluor 488 (1:1000, R37118, Thermo Fisher Scientific, USA)

- Donkey anti-Rabbit IgG (H + L) Alexa Fluor 568 (1:1000, A10042, Thermo Fisher Scientific, USA)
- Donkey anti-Rabbit IgG (H + L) Alexa Fluor 647 (1:1000, A12379, Thermo Fisher Scientific, USA)

Primary:

- Alexa Fluor 647 Phalloidin (1:500, A12379,Thermo Fisher Scientific, USA)
- AP2$\alpha$ (1:100, sc-12726, Santa Cruz Biotechnology, USA)
- Arrestin 3 (Cone Arrestin, 1:50, sc-54355, Santa Cruz Biotechnology, USA)
- Brn-3b (1:50, sc-31989, Santa Cruz Biotechnology, USA)
- CHX10 (1:200, sc-21690, Santa Cruz Biotechnology, USA)
- CRALBP (1:250, ab15051, Abcam, USA)
- EEA1 (1:500, 14-9114-80, eBioscience, Thermo Fisher Scientific, USA)
- EZRIN (1:200, 3145S, Cell Signaling, USA)
- LAMP2 (1:50, sc18822, Santa Cruz Biotechnology, USA)
- Melanoma gp100 (1:100, ab787, Abcam, USA)
- MITF (1:500, X1405M, Exalpha Biologicals, USA)
- PAX6(1:100, PRB-278P-100, Covance, USA)
- PKC$\alpha$ (1:500, sc-208, Santa Cruz Biotechnology, USA)
- Rhodopsin (1:200, sc-57432, Santa Cruz Biotechnology, USA)
- ROM1 (1:200, 21984–1-AP, Proteintech, USA)
- RPE65 (1:100, ab78036 Abcam, USA)
- Anti-Opsin, blue (1:200, AB5407, Merck Millipore, USA)
- ZO-1 (1:100, 61–7300, Thermo Fisher Scientific, USA)

## TUNEL assay

TUNEL Assay (Click-iT TUNEL Alexa Fluor 488 Imaging Assay; Thermo Fisher Scientific, USA) was performed according to the manufacturer's manual.

## Gene expression analysis using Fluidigm qRT-PCR

Total RNA isolation and gene expression analysis was performed as previously described (*Raab et al., 2017*). For quantification of the gene expression of the genes of interest, Taqman assays were purchased from Thermo Fisher Scientific, USA.

## Calcium imaging

RoCs were incubated overnight with BRDM containing 9-cis-Retinal (Sigma-Aldrich, USA), 0.27 µM Fura-2-AM and 0.1% pluronic acid (Invitrogen, USA) at 37°C and 5% $CO_2$. Afterwards, the RoCs were perfused with BRDM (5 ml) to wash out the excess dye. Ratiometric calcium-imaging recordings were performed utilizing an upright fluorescence microscope (BX50WI, Olympus, Germany) equipped with a 40x water immersion objective (LUMPlan FL, 40X/0.80W, $\infty$/0, Olympus), a polychromator (VisiChrome, Till Photonics, Germany) and a CCD camera (RETIGA-R1, 1360 × 1024 pixel, 16 bit). During the calcium-imaging recordings, stacks (single-plane two-channel) of the Fura-2 fluorescence at the focal plane of the ROs photoreceptors were acquired at 10 Hz ($\lambda_{exc}$ = 340 and 380 nm; Olympus U-MNU filter set, 30 milliseconds exposure time, 8-pixel binning) using the VisiView software (Till Photonics, Germany). The calcium-imaging ratio-stacks were generated by dividing the fluorescence images recorded at the excitation wavelengths of F340 and F380 (ImageJ, RatioPlus, https://imagej.nih.gov/). To detect the calcium signals in the RoCs, fluorescent-labeled cells were manually encircled by regions of interest (ROIs) and the obtained ROIs coordinates were used to extract corresponding calcium traces from the ratio-stacks. Average frames of pre- and post-stimulus frames substituted ratio frames during the light stimulation period (using the ImageJ building function Z-projection ('Average intensity').

## Fluorescence intensity quantification

The fluorescence intensity of the propidium iodid signal was quantified using ImageJ (https://imagej.nih.gov/) before and after PI labeling using ROI selection and mean intensity pixel values. Signal intensities of images taken before PI labeling were considered as background and subtracted from

the measured PI values. The mean PI fluorescence intensity of only the RO in the RoC ($I_{RO}$) was calculated via:

$$I_{RO} = I_{RO+RPE} - I_{RPE}$$

where by $I_{RO+RPE}$ is the mean PI signal intensity in the RO area ($A_{RO}$) from both RO as well as RPE and $I_{RPE}$ is the mean PI signal intensity of solely the RPE. $I_{RPE}$ was thereby calculated via:

$$I_{RPE} = \frac{I_{RoC} \times A_{RoC} - I_{RO+RPE} \times A_{RO}}{A_{RoC} - A_{RO}}$$

with $I_{RoC}$ representing the mean PI intensity of the entire RoC area ($A_{RoC}$).

## Microscopy
All microscopic images were as indicated in the individual panels either taken by an Imager.M2 Apotome1 (Carl Zeiss, Germany), LSM 710 Confocal microscope (Carl Zeiss, Germany) or by the EVOS FL Imaging System.

## Simulation of the fluidic transport processes
The free and porous fluid flow, as well as the transport of diluted species, was modeled according to work previously described (*Loskill et al., 2017*). Briefly, we created a simplified model of the RoC consisting of the media channel and the four tissue chambers, each with a diameter of 2 mm and a height of 1 mm. The porous PET membrane, between the media channel and tissue chambers, was modeled with a thickness of 10 µm. The incompressible stationary free fluid flow was modeled by the Navier-Stokes equation with the properties of water (dynamic viscosity µ = $1{\times}10^{-3}$ m$^2$/s, density ρ = 1000 kg/m$^3$) and a flow rate of 20 µl/h. Fluid flow from the media channel through the isoporous membrane into the tissue channel was modeled using Darcy's law (porosity = 0.056, hydraulic permeability κ = $1.45{\times}10^{-14}$ m$^2$). The transport of diluted species was described by the time-dependent convection-diffusion with a diffusion coefficient $1 \times 10^{-9}$ m$^2$/s and an initial concentration of 1 mol/m$^3$.

## Statistical analysis
To analyze differences between samples conditions, the two-sided student's t-test (*Figure 4g*), the one-way-ANOVA with a Bonferroni post-hoc test (*Figure 6h*), the one-way-ANOVA with a Dunnet post-hoc test (*Figure 8b,f,g*) or a two-way-ANOVA with a Bonferroni post-hoc test (*Figure 8h*) was used. Statistical analysis was performed with GraphPad Prism 7.04 (San Diego, California). Data are presented as mean ± S.E.M. p-value is indicated in the respective graphs.

## Data availability
The authors declare that the main data supporting the findings of this study are available within the article and its Supplementary Information files.

## Acknowledgements
This work was supported in part by the Fraunhofer-Gesellschaft Internal program Attract 601543 (PL), has received funding from the European Union's Horizon 2020 research and innovation programme under grant agreement No 766884 (PL), the DFG (DFG LI 2044/4-1 and DFG LI 2044/5-1 to SL, SCHE701/14-1 to KSL), the Ministry of Science, Research and the Arts of Baden-Württemberg (Az: 7542.2-501-1/13/6 to PL and Az: 33–729.55-3/214-8 to KS-L), the NC3Rs CrackIT scheme (PL and SL) and the Hector Fellow Academy grant to WH. The authors thank Dr. Murat Ayhan for advice on the statistical analysis.

## Additional information

### Funding

| Funder | Author |
| --- | --- |
| Fraunhofer-Gesellschaft | Peter Loskill |
| Deutsche Forschungsgemeinschaft | Katja Schenke-Layland<br>Stefan Liebau |
| Horizon 2020 Framework Programme | Peter Loskill |
| Ministerium für Wissenschaft, Forschung und Kunst Baden-Württemberg | Katja Schenke-Layland<br>Peter Loskill |
| National Centre for the Replacement, Refinement and Reduction of Animals in Research | Stefan Liebau<br>Peter Loskill |
| Hector Fellow Academy | Wadood Haq |

The funders had no role in study design, data collection and interpretation, or the decision to submit the work for publication.

### Author contributions

Kevin Achberger, Christopher Probst, Jasmin Haderspeck, Investigation, Methodology, Writing—original draft, Writing—review and editing; Sylvia Bolz, Wadood Haq, Investigation, Methodology; Julia Rogal, Johanna Chuchuy, Marina Nikolova, Virginia Cora, Lena Antkowiak, Investigation; Nian Shen, Katja Schenke-Layland, Marius Ueffing, Conceptualization, Writing—review and editing; Stefan Liebau, Peter Loskill, Conceptualization, Funding acquisition, Methodology, Writing—original draft, Project administration, Writing—review and editing

### Author ORCIDs

Kevin Achberger ⓘ https://orcid.org/0000-0001-5706-6201
Christopher Probst ⓘ https://orcid.org/0000-0002-1380-0117
Peter Loskill ⓘ https://orcid.org/0000-0002-5000-0581

### Decision letter and Author response

Decision letter https://doi.org/10.7554/eLife.46188.033
Author response https://doi.org/10.7554/eLife.46188.034

## Additional files

### Supplementary files
• Transparent reporting form
DOI: https://doi.org/10.7554/eLife.46188.031

### Data availability

The authors declare that the main data supporting the findings of this study are available within the article and its supplementary information files.

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
