## [Decision Letter]

Thank you for submitting your article "Merging organoid and organ-on-a-chip technology to generate complex multi-layer tissue models in a human Retina-on-a-Chip platform" for consideration by *eLife*. Your article has been reviewed by three peer reviewers, including Milica Radisic as the Reviewing Editor and Reviewer #1, and the evaluation has been overseen by a Reviewing Editor and Marianne Bronner as the Senior Editor.

The reviewers have discussed the reviews with one another and the Reviewing Editor has drafted this decision to help you prepare a revised submission.

Summary:

This paper describes a development of a retina organ-on-a-chip model. The paper is at a very forefront of the organ on the chip field. Retina is especially complex to model due to the large number of different cell types involved and complex structural organization. This paper describes integrating more than seven different cell types derived from human induced pluripotent stem cells in a vasculature like perfusion.

Importantly, the authors combined organoids with organ-on-a-chip engineering. This represents a high degree of novelty as it is not commonly done. They combined hiPSC derived retinal organoids with hiPSC-derived retinal pigment epithelial cells in a PDMS-based retina-on-a-chip device (RoC). Each device had four identical micro-tissues connected via a microchannel combining two transparent and biocompatible polymer layers.

The authors observed enhanced inner and outer segment formation and preservation, and they demonstrated the utility of this platform for drug testing. They demonstrated that retinal organoids expressed markers of major retinal cells such as: ganglion cells, bipolar cells, horizontal cells, amacrine cells, Müller glia and photoreceptors.

Overall this paper is strong in terms of the biological assessment as well as the engineering aspects related to device fabrication.

Essential revisions:

The reviewers agree that this manuscript is interesting and novel enough to warrant further consideration in *eLife*. The reviewers also agree that the authors should show the importance of perfusion in this system and demonstrate how it is superior to a transwell, which would form an important control. The reviewers agree this issue should be addressed experimentally. It is also necessary to perform a statistical review with an experienced statistician. The remaining small comments could be answered through appropriate explanation with references. Please see more detailed explanations of these comments in the reviews below.

Specific points:

Clarification on how long the entire structure is stable for and is there further remodelling of the structure with prolonged time in culture?

The phagocytic activity described in the paper is an inherent property of RPEs, and commercial assays exist to measure this function in mono-culture of RPEs using isolated, fluorescently labeled POS. Reproducing this function in the presented model system does not fully support the claim that RPEs interact with the retina organoids.

The acellular perfusion channel described in this model is not at all representative of the vasculature in the retina that is structurally and functionally integrated with other cell/tissue types in a complex 3D environment to maintain homeostasis. The configuration of the system may be used to simulate choroidal vessels but the channel used in this study is significantly divergent from choriocapillaris in vivo.

One potential benefit of perfusion is to maintain paracrine gradient between the retinal organoids and the epithelium. But without the transwell control, it's not clear how significant is this benefit in these experiments. For example, in Figure 5H, where would the condition with the co-culture of organoids and epithelium in a transwell setting be?

In the fluorescent intensity section, the authors should write out the formulas in the proper format.

The fourth sentence in the Abstract might need to be reworded.

*Reviewer #1:*

This paper describes a development of a retina organ-on-a-chip model. The paper is at a very forefront of the organ on the chip field. Retina is especially complex to model due to the large number of different cell types involved and complex structural organization. This paper describes integrating more than seven different cell types derived from human induced pluripotent stem cells in a vasculature like perfusion.

Importantly, the authors combined organoids with organ-on-a-chip engineering. This represents a high degree of novelty as it is not commonly done. They combined hiPSC derived retinal organoids with hiPSC-derived retinal pigment epithelial cells in a PDMS-based retina-on-a-chip device (RoC). Each device had four identical micro-tissues connected via a microchannel combining two transparent and biocompatible polymer layers.

The authors observed enhanced inner and outer segment formation and preservation, and they demonstrated the utility of this platform for drug testing. They demonstrated that retinal organoids expressed markers of major retinal cells such as: ganglion cells, bipolar cells, horizontal cells, amacrine cells, Müller glia and photoreceptors.

Overall this paper is strong in terms of the biological assessment as well as the engineering aspects related to device fabrication.

I would appreciate clarification on how long the entire structure is stable for and is there further remodelling of the structure with prolonged time in culture?

*Reviewer #2:*

This study is interesting and significant in that it is the first to demonstrate the incorporation of stem cell-derived organoids into RPE culture to model the retina, especially the photoreceptor cells which have proven challenging to recapitulate in vitro. Despite this positive aspect, the bulk of data presented in the paper are very preliminary and fail to substantiate the main claims made by the authors. First off, one major concern is that this system is presented as a vascularized retina model, which is completely misleading. The acellular perfusion channel described in this model is not at all representative of the vasculature in the retina that is structurally and functionally integrated with other cell/tissue types in a complex 3D environment to maintain homeostasis. The configuration of the system may be used to simulate choroidal vessels but the channel used in this study is significantly divergent from choriocapillaris in vivo.

The absence of data demonstrating the functionality and benefit of the perfusion channel is another major issue. In the introduction, the lack of vascularization in existing model systems is highlighted as one of the major problems that have motivated this study, but the presented results do not show whether and how the channel structure in the bottom layer serves to make this model superior and more physiological. Control data are nowhere to be found in the manuscript that compare the outcome of cell culture in the presence or absence of flow. On a related note, another critical question that remains unanswered is, is a chip-based system really necessary for this study? As discussed by the authors, MPS and organ-chip models offer numerous novel capabilities (e.g., precise control over the soluble and ECM microenvironment, incorporation and spatial patterning of different cell types, controlled delivery and application of external factors, etc.) not achievable in traditional in vitro models. It is highly questionable whether this study leverages any of these capabilities and whether this system is advantageous over conventional Transwell-based in vitro models. It would be readily possible to establish a similar co-culture model in Transwell inserts and conduct the same assays.

The claim that this model is the first to recapitulate the interaction between photoreceptor cells and RPEs also raises significant concerns. The phagocytic activity described in the paper is an inherent property of RPEs, and commercial assays exist to measure this function in mono-culture of RPEs using isolated, fluorescently labeled POS. Reproducing this function in the presented model system does not fully support the claim that RPEs interact with the retina organoids. The method of analysis used in this study is also problematic as poor resolution of imaging makes it difficult to interpret the data. It is understandable that the configuration of the device, especially the thickness of the bottom layer, presents challenges to high-resolution imaging analysis. If this really was the case, the poor imaging data seem to suggest the limitation of this chip-based model.

*Reviewer #3:*

The manuscript by Loskill et al., presents a Retinal-on-a-Chip that combines hiPSC-derived retinal organoids composed of all known major retinal subtypes and a monolayer of retinal pigment epithelial in a membrane-based microfluidic device. The paper provided through characterization on both retinal organoids and the retinal epithelium. When integrated, the study clearly demonstrated the functional indigestion of segment structures by the retinal pigment epithelium, which is a functional hallmark of the visual cycle. Furthermore, the Authors demonstrated pharmaceutical testing with an anti-malaria drug and an antibiotic and indicated that the co-culture condition could provide a protective role in the pharmaceutical response. Overall, I think the work is impressive, novel, and would be of great interests to the community. The data are well-presented and the discussions are very extensive and through. My only concern is regarding the utility of perfusion in the existing model. Although the experiments were done on a microfluidic platform, it seems all the experiment could also be performed in a standard transwell to achieve the same result. One potential benefit of perfusion is to maintain paracrine gradient between the retinal organoids and the epithelium. But without the transwell control, it's not clear how significant is this benefit in these experiments. For example, in Figure 5H, where would the condition with the co-culture of organoids and epithelium in a transwell setting be? I certainly see the potential utility of this model, especially by incorporating vasculature and perfusion in the future, which the authors have mentioned. Considering the potential of this platform and the biological finding presented, I think this paper can be accepted for publication.

---

## [Author Response]

Essential revisions:The reviewers agree that this manuscript is interesting and novel enough to warrant further consideration in eLife. The reviewers also agree that the authors should show the importance of perfusion in this system and demonstrate how it is superior to a transwell, which would form an important control. The reviewers agree this issue should be addressed experimentally.

Concerning the benefits of perfusion and the advantage over a transwell model, the authors would like to stress three important points:

1) To the author's knowledge, no reports of a comparable transwell model exist in scientific literature, to date. Initially, when starting to tackle the issue of a lack of physiological retina models, we considered a variety of possible approaches including a transwell model. However, there are no standard inserts fitting the size of ROs and the generation of defined spacings between ROs and RPE would have not been possible most likely. Moreover, since we did not see any advantages in a transwell style model but only disadvantages such as the missing perfusion, we decided to develop a microfluidic-based solution. In our view, it would be extremely unusual to now develop an additional entirely new system (transwell model), as a control for our introduced novel system, which we did compare to the gold standard dish cultures.

2) Perfused culture systems have the general advantage of stable and constant culture conditions. In static (transwell) dish cultures, the levels of nutrients and compounds are constantly decreasing and the levels of metabolized and secreted products are constantly increasing in-between media exchanges, leading to sawtooth-like concentration profiles and artificial periodic dynamics. In contrast, microfluidic perfusion provides a constant transport of fresh nutrients and compounds to the cells and of metabolites and secreted factors away from the tissue. This leads not only to physiological conditions (hence, “vasculature-like”) but also generates stable & constant conditions.

3) Perfusion enables the possibility to generate temporally very controlled exposures to compounds as well as to sample secreted factors in a time-resolved manner. Moreover, it paves the way for future connection to multi-organ-systems that circulate media between different platforms.

To underline these points, we conducted proof of concept experiments, which we describe in subsection “Physiological secretion kinetics into the vasculature-mimicking channels” and the new Figure 5 and Figure 6—supplement figure 1 as well as Video 2 and Video 3. We monitored the kinetics of the basal VEGF secretion during the stimulation with TGF-ß1 and subsequent washout. Unlike in a static (transwell) culture, the microfluidic RoC allows the continuous sampling of substances without removing/changing medium. We also conducted simulations as well as experiments highlighting the rapid and precisely controllable change of culture conditions and vasculature-like transport processes.

It is also necessary to perform a statistical review with an experienced statistician.

We consulted the experienced statistician Murat Ayhan, Ph.D. (Data Science for Vision Research, Institute for Ophthalmic Research, University of Tuebingen) for reviewing the statistics. Along with the reviewer’s suggestion, he suggested changing the statistical testing. It was changed to one-way-ANOVA with a Bonferroni Post-hoc Test (Figure 5H, Figure 7F,G) and a two-way-ANOVA (Figure 7H). The respective statistical paragraph in the method part was changed likewise.

Clarification on how long the entire structure is stable for and is there further remodelling of the structure with prolonged time in culture?

We conducted an additional experiment culturing the RoC for 21 days (Figure 4—figure supplement 1) to show the stability of the chip culture. In contrast to a drug-treated cell-death control (10 µg/ml Chloroquine), the tested chips showed low to no levels of Propidium Iodide positive cells.

The phagocytic activity described in the paper is an inherent property of RPEs, and commercial assays exist to measure this function in mono-culture of RPEs using isolated, fluorescently labeled POS. Reproducing this function in the presented model system does not fully support the claim that RPEs interact with the retina organoids.

We have to politely disagree with the reviewer. The fact that we observe a phagocytic activity of the RPE ingesting segments from the photoreceptors from the RO does clearly demonstrate the interaction between these two cell types. The phagocytic activity of RPE of isolated POS in mono-cultures is indeed well described. However, the here described phagocytosis of POS shed by the adjacent photoreceptor layer is a much more physiological process and an important aspect of the functionality of retinal tissue.

*The acellular perfusion channel described in this model is not at all representative of the vasculature in the retina that is structurally and functionally integrated with other cell/tissue types in a complex 3D environment to maintain homeostasis. The configuration of the system may be used to simulate choroidal vessels but the channel used in this study is significantly divergent from choriocapillaris* in vivo.

We do not claim to recapitulate the vasculature in the retina in its entirety. The model, however, enables a media perfusion similar to the in vivo blood perfusion through the choroidal vessels (hence referred to as “vasculature-like perfusion”). This constitutes an important functional aspect of the choriocapillaris in vivo.

One potential benefit of perfusion is to maintain paracrine gradient between the retinal organoids and the epithelium. But without the transwell control, it's not clear how significant is this benefit in these experiments. For example, in Figure 5H, where would the condition with the co-culture of organoids and epithelium in a transwell setting be?

We agree with the reviewer, the design of the platform leads to paracrine gradients as well as nutrient gradients, which resemble the physiological environments closer than RO dish cultures or other static culture conditions. However, as mentioned above, no comparable transwell-based retina model exists, which could serve as a reference.

In the fluorescent intensity section, the authors should write out the formulas in the proper format.

A proper format including a more detailed description has been added.

The fourth sentence in the Abstract might need to be reworded.

The sentence was rephrased.